# Combinatorial Therapies in Thyroid Cancer: An Overview of Preclinical and Clinical Progresses

**DOI:** 10.3390/cells9040830

**Published:** 2020-03-30

**Authors:** Gheysen Laetitia, Saussez Sven, Journe Fabrice

**Affiliations:** Laboratory of Human Anatomy and Experimental Oncology, Faculty of Medicine, Mons University, Avenue du Champ de Mars, 8, B7000 Mons, Belgium; Sven.SAUSSEZ@umons.ac.be (S.S.); Fabrice.JOURNE@umons.ac.be (J.F.)

**Keywords:** thyroid, cancer, targeted therapies, chemotherapies, drug combination, preclinical, clinical

## Abstract

Accounting for about 2% of cancers diagnosed worldwide, thyroid cancer has caused about 41,000 deaths in 2018. Despite significant progresses made in recent decades in the treatment of thyroid cancer, many resistances to current monotherapies are observed. In our complete review, we report all treatments that were tested in combination against thyroid cancer. Many preclinical studies investigating the effects of inhibitors of the MAPK and PI3K pathways highlighted the importance of mutations in such signaling pathways and their impacts on the subsequent efficacy of targeted therapies, thus reinforcing the need of more personalized therapeutic strategies. Our review also points out the multiple possibilities of combinatory strategies, particularly using therapies targeting proliferation, survival, angiogenesis, and in combination with conventional treatments such as chemotherapies. In any case, resistances to anticancer therapies always develop through the activation of alternative signaling pathways. Combinatory treatments aim to blockade such mechanisms, which are gradually decrypted, thus offering new perspectives for the future. The preclinical and clinical aspects of our review allow us to have a global opinion of the different therapeutic options currently evaluated in combination and to be aware about new perspectives of treatment of thyroid cancer.

## 1. Introduction

Thyroid cancer records approximately 2% of total cancers diagnosed worldwide and is the most common endocrine cancer [1,2]. In 2018, thyroid cancer was responsible for 567,233 cases worldwide, ranking in 9th place for incidence. Approximately 41,000 deaths were registered during the same period. The incidence of thyroid cancer has been increasing in many countries for the last 30 years, probably due to the increased diagnosis of papillary thyroid cancer due to improvements in the detection methods. However, it is important to note that the overdiagnosis is estimated to account for 50–90% of newly diagnosed cases in some regions in the world, such as South Korea, the United States, Japan, and some European countries including Italy and France [3,4].

The main known risk factors are the exposure to ionizing radiation especially during childhood, a history of thyroid disease (goiter, thyroiditis, or adenoma), an exposure to certain environmental pollutants or an iodine deficiency [3,5]. Obesity has been broadly studied for numerous cancers and can be associated with approximately 5% of all cancers. A correlation has been reported between the risk of developing thyroid cancer and a high body mass index (BMI) for both men and woman [5,6,7]. Moreover, Liu et al. demonstrated that a high BMI not only affects the risk of developing thyroid cancer, but also influences the risk of extrathyroidal invasion of cancer cells [6].

Thyroid cancer affects women three times more frequently than men, as supported by a population-based study in 29 European countries reporting that 76% of patients were women. This percentage can reach more than 80% in some countries, such as Portugal, Malta, and most Eastern European countries [8]. Regarding the recent worldwide studies published in 2018, 130,889 men and 436,344 women were diagnosed for thyroid cancer, confirming the highest incidence in females [3]. The reasons for explaining this disparity are not yet completely established. Many authors have studied the possible relationship between female hormonal and reproductive processes, such as pregnancy, menstrual cycle, menopause, and hormone replacement therapy (HRT). However, none of these criteria were consistently associated with a higher risk of developing thyroid cancer [5,9]. According to in vitro experiments led by Rajoria et al., the presence of a functional estrogen receptor (ER) can be implicated in the cellular process leading to enhanced mitogenic, migratory, and invasive properties of the thyroid cells [10].

Thyroid cancers can be categorized into different histological types, which are the papillary, the follicular, the Hurthle cell and the anaplastic carcinomas, all derived from the follicular cells, and the medullary thyroid form derived from the parafollicular or C cells. The well-differentiated forms grouping the papillary (PTC), the follicular (FTC), and the Hurthle cell (HCC) carcinomas are characterized by the maintaining of the thyroid cell organization. It is important to note that gender disparity is mostly observed for PTC and FTC [8], which represent almost 90% of thyroid cancers diagnosed with a predominance of the papillary one (80%) [4,11]. The high rate of PTC is maintained even after the reclassification of non-invasive encapsulated follicular variant PTC (EFVPTC) as non-invasive follicular thyroid neoplasms with papillary-like nuclear features (NIFTP) [12]. Of note, HCC is only detected in about 3% of thyroid cancer patients and it is not considered a subtype of FTC in the 2017 WHO Classification of Tumors of Endocrine Organs. [13]. Patients with these differentiated cancers have a good prognosis with a 5-year survival rate higher than 95% [14,15]. In case of Hurthle cell carcinoma, the prognosis is often encouraging and after adequate surgical treatment recurrences are rare [16]. Unlike Hurtle cell carcinoma, the development of local recurrence and/or metastasis in the cases of resistance to the radioiodine (RAI) treatment can affect between 5% and 15% of patients with differentiated thyroid cancer (DTC) from mainly follicular origin and seriously impacts the prognosis. Nearly 2/3 of these patients will become RAI-refractory (RR-DTC) with a significant reduction of the mean life expectancy of 3–5 years [17,18]. In this clinical context, four categories of patients are distinguished: (1) patients with metastatic disease that do not take up RAI at the time of the initial treatment, (2) patients whose tumors lose the ability to take up ^131^I after previous evidence of uptake, (3) patients with ^131^I uptake retained in some lesions but not in others, and finally (4) patients with metastatic disease that progresses despite significant RAI uptake in their metastases and following courses of adequate radioiodine treatment [19]. Patients with radioiodine refractory thyroid cancer are characterized by advanced age (median age >60 years), a poorly differentiated follicular thyroid cancer, a large metastasis burden, and a high ^18^FDG uptake on a PET scan. A metastatic or locally advanced thyroid cancer is considered as radioiodine refractory if it has at least one metastatic site without any radioiodine uptake or significant disease progression during the year after a radioiodine administration [20]. Indeed, nearly half of the persistent/recurrent or metastatic lesions lose the ability to take up radioiodine de novo or through a dedifferentiation process leading to mechanisms of resistance [21].

Medullary thyroid cancer (MTC) is a well-differentiated neuroendocrine tumor, which originates from the parafollicular cells (calcitonin-producing C-cells) and accounts for 5–10% of diagnosed thyroid cancers [22]. Approximately 25% of MTC is hereditary due to germline mutations [23]. Total thyroidectomy and adequate neck dissection are efficient in patients with such localized diseases. However, MTC still remains a challenge for physicians because it frequently persists after surgery, grows slowly, and metastasizes to neck and mediastinum lymph nodes, as well as to the bones, the liver and, more rarely, other organs. Moreover, there is currently a lack of available therapies with a proven survival benefit for MTC patients [24].

Anaplastic thyroid carcinoma (ATC) is one of the most aggressive types of any cancer. It represents only 1–2% of all thyroid cancers, but the mortality rate is close to 100% [25]. The median survival rate is not higher than 6 months after diagnosis [26,27,28]. Patients with ATC are, most of the time, more than 50 years old. The progression of the disease is very fast, with the size of the tumor doubling within 1 week [29]. As described by the American Joint Committee on Cancer (AJCC), an advanced stage disease at diagnosis is the most frequently observed and is therefore characterized as a stage IV disease [30]. ATC develops as a large mass in the neck region, causing mechanical compression on neighboring organs, such as the trachea, the esophagus, and the recurrent laryngeal nerve. Patients can thus suffer from dysphagia, which can lead to anorexia and weight loss, airway obstruction, hoarseness, and cervical pain. The reason for the difficulty in operating ATC patients is the presence of vital vessels and nerves near the tumor [27,29,31,32]. The association of these symptoms and complications from therapies leads to a rapid death [27]. The size of the tumor is frequently superior to 5 cm, as observed in many studies [33,34,35]. ATC is also known to frequently metastasize in sites, such as the lungs (80%), the bones (6–16%), and the brain (5–13%) [25,31,34]. The origins of the development of the anaplastic form still today remains unclear but more and more evidences about histologic observations and genetic alterations studies support the hypothesis according to which the anaplastic form could derive from a dedifferentiation of the papillary form. The presence of coexisting PTC, FTC, or PDTC is a argument of favor of the development of ATC from a well-differentiated thyroid carcinoma [25]. The study of BRAF mutation, documented as the most common mutation in PTC, reveals a high prevalence of this mutation found in ATC [36]. As developed in some studies, the accumulation of genetic alterations seems to be a mechanism of ATC development [37,38].

The aim of this review is to report the current worldwide knowledge at preclinical and clinical levels, leading to the present and future strategies to overcome resistance mechanisms by treating patients diagnosed with a thyroid carcinoma with combinatorial therapies.

## 2. Preclinical Studies Combining Treatments in Thyroid Cancer Cells

Many preclinical studies have been conducted in order to evaluate the effect of various inhibitors on a panel of thyroid cancer cells. Data reporting the combination of such drugs are reported in Table 1.

### 2.1. Targeting MAPK and PI3K Signaling Pathways in Thyroid Cancer Cells

In the context of cancer, MAPK and PI3K signaling pathways are parts of the most successfully studied pathways [39]. These pathways include various gene alterations (mutation, amplification, and deletion) that may contribute to the carcinogenesis and that this may be targeted for therapies.

Within the MAPK pathway, the RAF proteins serve as master regulators of a myriad of downstream signaling cascades. They control essential proteins for cell growth, proliferation, migration, apoptosis, cytoskeletal integrity, survival, and differentiation [40]. The serine/threonine RAF kinases induce the phosphorylation of the downstream MEK1 and MEK2 kinases, which in turn activate the ERK1 and ERK2 kinases [40,41].

Approximately 7–9% of all human cancers harbor BRAF mutations, which is generally a substitution of glutamic acid for valine at position 600 (BRAFV600E). Consequently, this mutation has the constitutive activation of BRAF kinases and, therefore, the MAPK signaling pathway favoring the development of biologically and clinically aggressive forms of solid malignancies, which are moreover frequently resistant to conventional anticancer therapies [41,42]. Thyroid cancer may be classified among cancers bearing this mutation the most frequently with a prevalence of 36–53% after melanomas (40–70%) [40,42,43]. However, the prevalence of BRAF mutation in thyroid cancer varies according to the form of thyroid cancer. Papillary and anaplastic thyroid carcinomas are often affected, contrary to follicular and medullary thyroid malignancies, which are not [43]. In PTC, a direct association between BRAF mutation and aggressive clinicopathologic features exists, such as extrathyroidal invasion, large tumor size, lymph node metastasis involvement, and rapid occurrence of recurrences [41,44]. It is also important to note that the frequency of BRAF mutation in ATC varies according to the region with a higher prevalence in Japan compared to Western countries [45].

The PI3K/AKT signaling pathway was discovered in the 1980s [46]. It plays major roles in several cellular events including growth, proliferation, apoptosis, metabolism, motility, and resistance to treatment. The activation of this crucial pathway is implicated in many mechanisms leading to carcinogenesis [41]. The active form of AKT (phosphorylated form) is considered as an antiapoptotic regulator due to its numerous interactions with proteins implicated in the apoptosis process such as Bad, caspases 3 and 9, and Bax. PI3K/AKT is also implicated in the regulation of the expression of HIF1α, which, once upregulated, can induce the synthesis of VEGF broadly implicated in the angiogenesis process [47].

The PI3K/AKT signaling pathway is closely associated to other crucial pathways such as Wnt-β catenin, FOXO3, NFκB, or MAPK also implicated in the tumorigenesis process [47]. RAS mutation can activate the PI3K/AKT pathway. Therefore, activation of the MAPK pathway and/or PI3K/AKT pathway implicated in the development of an aggressive form of PTC can then turn into an ATC by dedifferentiation [48]. The concomitant inactivation of the tumor suppressor PTEN and the activation of RAS oncogenes have been described to play a role in thyroid carcinogenesis [41]. PTEN is known to antagonize the PI3K/AKT pathway and its loss stimulates this pathway. Around 12% of ATC exhibit mutated or deleted PTEN genes leading to an overactivation of the PI3K/AKT pathway and a more aggressive tumor [39].

This complex network of connections between these crucial cellular signaling pathways seems to be the key of the development and recurrence of thyroid carcinomas. This explains that, for approximately 10 years, MAPK and PI3K/AKT are the targets of many treatment-type research. The study of new combinations of drugs able to escape to the resistance mechanisms developed by the tumors exposed to single targeted agents is now a major concern for the treatment of thyroid carcinomas.

#### 2.1.1. MAPK and PI3K Inhibitors Used as Single Agents

A huge quantity of new targeted treatments is available and tested in clinical trials due to the increasing knowledge on the molecular alterations in cancer. However, after several years of craze and hopes with monotherapies, scientists have been confronted with major and recurrent problems. Monotherapies are often associated with resistance, recurrence, and sometimes worsening of the disease [49,50].

Approximately 1/5 of patients harboring activating mutations in BRAF develop intrinsic resistance and do not respond to the BRAF inhibitor [51]. Vemurafenib is a mutant BRAF inhibitor and it is one of these molecules that are failing in clinical assays as monotherapy [52]. In vitro studies have been conducted to elucidate resistance mechanisms but reasons explaining the higher resistance observed of thyroid BRAFV600E mutant thyroid cancer in comparison to melanoma to BRAF inhibitor remains unclear [51]. Compensatory/alternative mechanisms have been highlighted to promote such resistance, bypassing pharmacologic inhibition of BRAFV600E via the activation of intracellular signaling pathways, leading to the reactivation and phosphorylation of ERK1/2, NRAS mutation, mutant BRAF amplification, or alternative splicing [51]. Among them, aberrant autocrine loops were developed through the overexpression of the HER3 receptor [53]. Besides, Montero-Conde’s team demonstrated that thyroid tumor cells harboring the BRAFV600E can overexpress the HER signaling pathway when treated with BRAF or MEK inhibitors, reactivating ERK and/or AKT [54]. This feedback reactivating the HER signaling also causes the activation of JAK/STAT signaling pathways, as well as SRC kinase, which ultimately causes treatment failure [55]. Other mechanisms, which are associated to eIF4F complex formation, are implicated in the reactivation of ERK1/2 signaling or in persistent activation of p-mTOR and p-S6 ribosomal proteins signaling [56].

The failed attempts of vemurafenib used as monotherapy have been studied in the field of thyroid cancer and has already been well-documented in melanoma and colorectal cancer. One critical aspect to improve cancer therapy is not only to inhibit the primary oncogenic pathway reducing by this way the proliferation but simultaneously prevent functional redundancies and pathways crosstalk that facilitates survival of cancer cell populations, rendering tumors resistant to treatment [51].

#### 2.1.2. MAPK and PI3K Inhibitors Used in Combination

New hopes are occurring with the combined inhibition of both MAPK and PI3K pathways. This strategy is the object of many preclinical studies. Most of them demonstrate synergistic action on cancer cell lines and/or in the murine model as described below. The impact on cellular pathways is multiplied due to the central role of such pathways (Figure 1). Most importantly, the purpose of this combination is to achieve stable and durable responses and, in this way, to oppose resistance mechanism problems in thyroid cancer treatment.

##### Combination to MAPK/MEK Inhibitors

As previously mentioned, most aggressive thyroid cancers often express the BRAF^V600E^ mutation. Investigations on thyroid cancer cells harboring such mutations have been conducted, combining the MEK inhibitor selumetinib and the proteasome inhibitor bortezomib. The MAPK pathway interacts not only with the PI3K one but also with the NFκB cascade, indirectly targeted by the bortezomib. The interaction between these two molecules resulted in a synergistic antiproliferative effect in cancer cells. Most importantly, the doses of the two drugs were reduced in the animal model, while the synergistic antitumour effects were preserved. In vitro assays studying the invasion suggested that the treatment could also play a role in the metastatic process [57]. The redifferentiation of cells harboring the BRAF^V600E^ mutation was reported to be associated with an increase of radioiodine uptake and was evaluated with the combination of BRAF/MEK inhibitors (dabrafenib/selumetinib) and HER1/2 inhibitor (lapatinib). Additionally, the decrease of ^18^F-FDG uptake seems to be attributed, in part, to the changes in expression of GLUT1 and GLUT3. After the combination treatment, the expression levels of GLUT1 in DTC (differentiated thyroid cancer) cells strongly decreased, while the expression of GLUT3 remained unchanged. However, the rationale for in vivo and clinical studies is provided by the promising combination of dabrafenib/selumetinib and lapatinib in conjunction with ^131^I therapy [58]. In another study, dabrafenib/selumetinib were combined with a histone deacetylase inhibitor (HDACi), the panobinostat. The effects of these molecules were only tested on cells harboring BRAF mutation. Panobinostat alone increased iodine-metabolizing genes expression, as well as promoted radioiodine uptake, induced cell toxicity, and suppressed GLUT1 expression. However, combined treatment displayed a more robust BRAF^V600E^-dependent redifferentiation effect than panobinostat alone due to an improved acetylation level of histones at the sodium-iodide symporter (NIS) promoter. However, these encouraging results should be further investigated with an in vivo model to confirm the robustness of these in vitro data [59].

Broadly studied to understand the physiologic mechanisms of the thyroid, the sodium iodide symporter (NIS) shows a crucial role in the radioiodine-refractory papillary thyroid cancer (RR-PTC). Indeed, as for ATC, RR-PTC is characterized by a poor prognosis. A major mechanism underlying the development of RR-PTC is the aberrant silencing of iodine metabolizing genes such as Nis, Tshr, Tg, and Tpo, which is a result of BRAF^V600E^ mutation induced activation of the MAPK pathway [21]. The histone H3 lysine 27 (H3K27) trimethylation modification (H3K27me3) leads to a decrease of gene expression through an enhancer of zeste homolog 2 (EZH2), which is a critical methyltransferase catalyzing H3K27 and an epigenetic mark for the maintenance of gene silencing. This hyper-trimethylation on H3K27 seems to be associated with cancer cell dedifferentiation and resistance to the BRAF inhibitor treatment and has been clinically found to be up regulated in poorly differentiated and anaplastic thyroid cancers. MAPK signal aberrant activation by BRAF^V600E^ has also been demonstrated to increase the level of H3K27me3 through increasing the expression of Ezh2 in thyroid cancer. Therefore, inhibiting the activity of EZH2 by specific inhibitors represents a potential direction of differentiation therapy. In this context, a study has been conducted to evaluate the differentiation efficacy of EZH2 inhibitor (tazemetostat) and to assess the impact on differentiation induced by the EZH2 inhibitor combined with a MAPK inhibitor (dabrafenib or selumetinib). The conclusions of this work are that tazemetostat promoted ^125^I uptake in PTC and the combination of the molecules induced a more robust expression of Nis and Tshr in BCPAP and K1 cells. The combination induced an inhibition of the expression and the activity of EZH2 yielding a strong reduction of the downstream H3K27me3, which play a role in the differentiation of mutant BRAF PTC cells. This combination including MAPK inhibitors and tazemetostat may be potentially translated into a novel differentiation therapeutic strategy [21].

Firstly developed to treat metastatic melanoma with the BRAFV600E mutation, the vemurafenib gave as a first step promising results before being confronted with resistance problems. One mechanism of acquired resistance is an uncontrolled activation of tyrosine kinase signaling alternate pathways. In the case of thyroid cancer, some resistances seem to be attributed to the activation of parallel signaling pathways, such as the EGFR signaling pathway. In this context, the inhibition of both EGFR and BRAF has been evaluated and results in superior responsiveness in comparison to the effect for the single agent. The blockade of these two pathways by vemurafenib and gefitinib, respectively, stopped the phosphorylation of EGFR and sustained the inhibition of ERK and AKT signaling abrogating compensatory mechanisms of cell survival [42]. The consequences of the mutant BRAF^V600E^ mutation are broadly studied in cancers, such as melanoma and thyroid cancer, and the mutational status of cell lines is frequently used to try to understand and explain the response mechanisms to treatments.

As described many times in the literature, targeting ERK seems to give promising results for both in vitro, in different thyroid cell lines, and in vivo, in various mouse models. This strategy was explored by combining the molecules vemurafenib and ponatinib (tyrosine kinase inhibitor) and comparing results obtained in cell lines harboring BRAF^V600E^ (8505C-BCPAP) versus BRAF^WT^ (THJ-16T-C643). Vemurafenib showed good synergism with ponatinib on cells mutant BRAF for the inhibition of cellular proliferation, colony formation, invasion, and migration. The apoptosis is more elevated by a significant increase in caspase 3/7 activity. The inhibition of ERK and MEK was also inhibited. On the contrary, the combination tested on BRAF^WT^ cell line showed mostly an antagonistic effect in proliferation assays, no changes in the phosphorylation of ERK and MEK and in apoptosis results reporting heterogeneity with no consistent increase of caspase activity after treatment. In in vivo 8505C orthotopic mouse model, the combination of vemurafenib with ponatinib significantly reduced tumor growth compared to the vehicle group control and single agent groups. Hematoxylin and eosin staining demonstrated a reduction of pulmonary metastasis in the vemurafenib treated group and combination treated group. Finally, assays on 8505C resistant to vemurafenib showed that the combination with ponatinib overcomes the acquired resistance [51].

The Bcl-2 family is also playing a crucial role in the apoptosis mechanism, which is often studied in cancer. In this context, a triple-drug combination associating ABT-737, PLX4720 (vemurafenib), and PD32590 was tested to inhibit Bcl-2 and BRAF, and to increase the proapoptotic proteins as Bim, while decreasing some antiapoptotic ones, such as MCL-1. The level of sensitivity of such a treatment in different cell lines suggests a possible dependency on the mutational status. However, an orthotopic mouse model reveals dramatic reductions in tumor volume and high rates of sustained apoptosis only associated with triple therapy [60]. The same MAP kinase inhibitor PLX4720 (vemurafenib) has been tested in the presence of the powerful SRC inhibitor dasatinib in vitro and in an immunocompetent orthotopic mouse model of anaplastic thyroid cancer. An in vitro test demonstrated that PLX4720 and dasatinib was limited by more than 90% the migratory capacity of TBP-3868, TBP-3743, TBPt-3403, and TBPt-3610R murine thyroid cell lines. The effect on the immune response has been also studied in vivo. As for the previous study described above, reduced tumor volume and increased immune cell infiltration, including cytotoxic T cells, B cells, and macrophages, were observed with the combination. These last results seem primordial in challenging the treatment of ATC and should deserve further investigations [61]. The antitumoral potential of the SRC inhibitor dasatinib has been also studied in the context of a combination with a MAPK inhibitor, trametinib. The efficacy of the combination therapy has been explored across a panel of thyroid cancer cell lines representing common oncogenic drivers (BRAF, RAS, and PIK3CA). The regulation of the expression of the PI3K pathway has been highlighted as crucial in the response to the treatment. In fact, they provided evidence to support a role for SRC in mediating activation of the PI3K pathway, and the ability to inhibit PI3K signaling correlates with sensitivity to combined SRC and MAPK pathway inhibition. They also, interestingly, mapped downstream phosphorylation of rpS6 as a key biomarker of response because of its powerful indicator of the effective reduction in MAPK and PI3K pathway activation [62].

A study conducted in 2013 has suggested the use of an AKT inhibitor, wortmannin, in combination with vemurafenib targeting mutant BRAF^V600E^ and with the anti-inflammatory agent apigenin. The apigenin acts as a disrupter of the mitochondrial membrane and promotes mitochondrial dysfunction causing cell death in FRO and 8505c cells in a time- and concentration-dependent manner. The use of wortmannin was justified by the resistance of phosphorylated AKT expression. Moreover, they demonstrated that the suppression of AKT potentiates the combined effect of apigenin and PLX4032 resulting in high rates of cytotoxicity [63].

Many investigations combining the BRAF inhibitor dabrafenib and the MEK inhibitor trametinib have led to FDA approval of these molecules for metastatic melanoma. This combination is currently tested in ongoing clinical trials for other metastatic cancers. The objectives are to clarify the resistance mechanism according to the differences in genetic alterations of cell lines, the level of VEGF secretion and/or the expression of EMT markers. Their first conclusion was that the inhibition of BRAF^V600E^ by dabrafenib could be effective against cancer cells harboring active alterations in both MAPK and PI3K/mTOR pathways. A study has investigated the effect of dabrafenib alone or combined with trametinib on four different cell lines harboring a mutational profile different form each other. They concluded that dabrafenib strongly inhibits the viability in BRAF mutated cells by demonstrating G0/G1-arrest via the downregulation of MEK/ERK phosphorylation. However, an upregulation of MEK phosphorylation was observed in RAS mutated cells after dabrafenib treatment causing a VEGF upregulation, but no evidence was found to correlate this fact to cellular proliferation. Trametinib inhibited the cellular viability to variable degrees in every cell line by downregulating ERK phosphorylation. Finally, the combination of dabrafenib + trametinib ended by a clear cytostatic effect in all the cells [64]. To overcome the mechanism of resistance to trametinib in ATC, the MEK inhibitor has been tested in combination with several molecules. Among them, JQ1 was selected for its efficient effect on cell invasion and metastasis development, which are the main concerns in ATC. JQ1 only acts spectacularly on tumor growth and cell proliferation by interfering with the BET protein family implicated in epigenetic mechanisms. JQ1 thus blocks the transcription of the MYC gene. It is important to note that the MYC protein has been studied in ATC and seems to be a critical oncogene associated to a poor patient prognosis when overexpressed in cancer. Thus, the in vivo combination of trametinib and JQ1, acting both on cell invasion and tumor growth, leads to a dramatic inhibition of tumor growth (>90%). Therefore, the combination of molecules targeting a large number of pathways from the transcription of genes to the activity of proteins seems to be a successful therapeutic strategy in cancer treatment and especially in ATC [65].

As often developed in the literature, the inhibition of the phosphorylation of the proteins ERK and MEK are crucial to decrease the tumor growth and invasion in thyroid cancer. However, one of the major limitations met with the MEK-ERK inhibitor is a severe toxicity in highly proliferative tissues such as skin and intestinal epithelial barrier but also a response limited in the time. Selenium has currently gained interest in thyroid cancer due to his potential properties to avoid toxicities in association with MAPK inhibitors. Others studies have suggested the possible implication of selenite in the stimulation of the immunity, the activation of natural killer cells, the inhibition of angiogenesis, and the enhancement of damaged DNA fragment repair but also the initiation of apoptosis in many cancers [66]. The U0126 MEK-ERK inhibitor has been tested on TPC1, 8505C, and hTori 3 cells in combination with sodium selenite. The molecule U0126 is a specific and noncompetitive inhibitor of both MEK1/2, which suppresses ERK phosphorylation and activation. The inhibitor decreased the growth thyroid cancer cell lines was observed while it did not significantly affect the growth of normal thyroid cells. They observed a stronger downregulation of ERK pathway with the addition of 5 µM of sodium selenite than without the selenite. The decrease of the expression of p90^RSK^ confirmed that sodium selenite indeed inhibited the ERK signaling pathway. This study showed encouraging adjuvant treatment efficiency to reduce MAPK inhibitor toxicity despite some study limitations. Indeed, only cell lines harboring the most common mutations in BRAF and the RET/PTCP1 rearrangement have been tested. Further investigations should be conducted in the cell line with RAS mutations and validated in animal models [66].

RDEA119 is a molecule with the advantage of an easy oral dosing and an excellent selectivity for MEK and has been associated with the mTOR inhibitor temsirolimus. The in vitro response to this combination was particularly excellent when both MAPK and PI3K/AKT pathways were mutated, highlighting the importance of mutational status in the choice of treatment. The cotreatment resulted in significant synergistic inhibition of the proliferation of thyroid cancer cells and the growth of xenograft thyroid tumors [67].

Today, the immune system is broadly studied in cancer treatment. More specifically, the interaction of programmed cell death-1 (PD-1) and its ligand (PDL-1) is the subject of many studies on all cancers combined. In the case of thyroid cancer, only a few articles discuss this subject despite the flourishing future of immunotherapy used in combination with targeted therapy. The immunocompetent murine model of anaplastic thyroid cancer has been tested with the combination of the PDL-1 antibody and the BRAF inhibitor PLX4720. The combination dramatically reduced tumor volume, compared to the use of molecules alone, and induced an intense CD8+ CTL (Cytotoxic T lymphocyte) infiltration and cytotoxicity, as demonstrated by immunohistochemistry analyses [68].

Furthermore, the β-galactoside-binding protein, plays a crucial role in diagnosing biomarkers and is also an activator of important signaling cascades [69,70]. Galectin-3 can bind RAS proteins with a preference for KRAS. The overexpression of galectin-3 is also known to promote neoplastic transformation by interacting with KRAS-GTP. These binding properties have been explored for treatment by using both inhibitors of the prenylation of RAS proteins (S-trans, transfarnesylthiosalicylic acid) and the galectin-3 (modified citrus pectin). The disruption of these interactions leads to the loss of the anchorage of RAS to the plasma membrane and consequently its degradation. Results showed inhibition of cell proliferation in vitro and a significant decrease of p53 levels was noticed. Tumor growth in vivo was also strongly diminished. These innovative findings could be clinically relevant and should lead to the development of new approaches for treating ATC patients [69].

##### Combination to PI3K Inhibitor

The molecule RAF265, originally developed for melanoma treatment, has a significant RET inhibitory activity, activity against wild-type and mutant BRAF, as well as an antiangiogenic activity by inhibiting VEGFR-2. Associated to the PI3K inhibitor BEZ-235, RAF265 causes a strong G1/G0 cell cycle arrest [71]. With the capacity of the molecule to target RET mutation, the potential of RAF265 to effectively treat the MTC form is important. In this context, combined with the PI3K inhibitor ZSTK474, RAF265 induced a strong cytotoxic effect and a strong inhibition of VEGFR-2 phosphorylation in TT cells harboring genetic alterations in RET [72].

Currently, it is well-known that mutations in p53 (loss of function) are crucial in the oncogenesis process and occurs with the highest frequency in ATC compared to other types of thyroid cancer. Few studies have previously demonstrated that concomitant PI3K activation and p53 mutation accelerated the development of drug resistance and the progression of thyroid cancer. In this context, the molecule PRIMA-1Met reactivates mutant p53 and has been already tested in melanoma, showing encouraging results in combination with the BRAF and MEK inhibitors [73,74]. Recently, the pan PI3K inhibitor NVP-BKM120, causing the suppression of PI3K downstream signaling, has been associated with the PRIMA-1Met and displayed synergistic antitumour effects both in vitro and in vivo thyroid cancer, thus opening the doors to a potential novel approach to the thyroid cancer treatment [75].

Regarding metastasis development, the combination of the AKT inhibitor MK-2206 and the silencing via shRNA of TGF-β1 in the ATC xenografts immunodeficient mouse model has been evaluated. Nevertheless, even if the inhibition of cell growth was significantly higher with the combination than with the molecules alone, it was concluded that the combination could not prevent the development of lung metastasis as expected [76]. The same AKT inhibitor MK-2206 has been also associated with the PDGFR inhibitor tyrphostin (AG1296). In this experiment, in vitro Boyden chamber assays showed promising results with an additive inhibition of cell invasion. In the future, in vivo experiments should be performed to confirm this positive result [77].

The epithelial–mesenchymal transition (EMT) is a key process for invasion mechanisms. An intranuclear accumulation of pAKT and nuclear exclusion of p27 to the cytoplasm could be directly associated with the invasiveness capacity of thyroid cancer cells and the acquisition of a mesenchymal phenotype. This observation suggests that AKT activation is involved in a direct induction of EMT. A hypothesis is that the combination of the BRAF and the c-Met inhibitors should reverse such EMT. The acquired resistance to BRAF inhibition promoted not only tumor progression and proliferation, but also migration and invasion of mutant BRAF thyroid cancer cells through the upregulated EMT induced by a c-Met-mediated AKT activation. This data opens the doors to additional investigations combining PI3K inhibitors and new molecules of targeted therapy to improve the treatment of thyroid cancer [78]. Recently, the PI3K/mTOR dual inhibitor omipalisib was associated with the CDK4/6 inhibitor palbociclib to overcome the resistance due to the accumulation of high levels of cyclin D1 and D3 in ATC. The combined treatment synergistically reduced cell proliferation and more crucially, the low-dose combination was dramatically effective in inhibiting tumor growth in a xenograft model. This therapeutic approach constitutes a highly promising strategy for the treatment of aggressive and resistant thyroid cancer [79].

##### Combination of MEK and PI3K Inhibitors

The cooperation between multiple acquired alterations, such as mutations in PI3KCA, TP53, and PTEN, can result in progression from well-differentiated thyroid cancers to non-differentiated ATC. By inhibiting both MAPK and PI3K pathways using the MEK inhibitor PD-325901 and the PI3K inhibitor GDC-0941, ElMokh et al. demonstrate the crucial role of these pathways to induce transformation and drive cancer cell proliferation. Most importantly, they conducted in vivo experiments and observed that MEK inhibition may induce a process of redifferentiation of thyroid cancer cells. This discovery plays in favor of the hypothesis that ATC could be a dedifferentiation from a pre-existent PTC. In this context, the MEK inhibitor AZD6244 (selumetinib) combined to the AKT inhibitor MK-2206 acted in synergy to inhibited thyroid cancer cell proliferation in lines harboring activating mutations in both pathways in opposition to cells harboring single or no mutation [80].

### 2.2. Targeting Angiogenesis in Thyroid Cancer

#### 2.2.1. VEGF Inhibitor Used as a Single Agent

In the context of thyroid cancer, a variety of targeted agents have been developed and tested in monotherapy. Among these molecules, a specific category of inhibitors such as tyrosine kinase receptor inhibitors has been broadly evaluated to inhibit angiogenesis. Tumor vascularization is a crucial process in cancer progression and vascular endothelial growth factor receptors (VEGFRs) were attractive targets for innovative therapies [50]. However, use in monotherapy, no study has demonstrated significant results.

For example, sorafenib (Nexavar^®^), initially developed by Bayer Pharmaceuticals as BAY 43-9006 in 2001, is a multikinase inhibitor that interferes with a myriad of signaling pathways, such as VEGFR-1, -2 and -3, BRAF, RET, cKIT, PDGFR, and Flt-3 (72–75). Sorafenib is approved by FDA for patients with RAI-refractory DTC as Lenvatinib. The current targets known for this molecule are VEGFR1-3, fibroblast growth factor receptors 1–4 (FGFR1-4), RET, c-kit, PDGFRa, and mast/stem cell growth factor receptor (SCFR) [81]. These molecules thus interact with many essential functions of cancer cells as proliferation, apoptosis, autophagy, and angiogenesis [75], but, alone, weak antiproliferative properties are observed for sorafenib and for lenvatinib, clinical studies discussed below showed adverse effects for >50% of patients included hypertension, diarrhea, or decreased appetite [81]. In this context, the strategy to combine these molecules to potentiate their properties with other molecules has been the object of preclinical studies described below.

#### 2.2.2. VEGF Inhibitor Used in Combination

Next, VEGF inhibitors have been evaluated as part of treatment combinations to synergize their antitumour effects. Even if the resistance mechanisms are still not deeply understood, combinations including sorafenib with other molecules try to overcome these major problems (Figure 2). Concerning thyroid cancer, different classes of molecules have been tested both in vitro and in vivo with promising results. Among them, histone deacetylase (HDAC) inhibitors, antimalaria drug, antidiabetic drug, curcumin, or autophagy inhibitor have been tested in combination with sorafenib [75,82,83,84].

Besides, RNA interference technology (siRNA) silencing AKT was combined with sorafenib resulting in a reduction of phosphorylation levels of both ERK and AKT. Cell viability was inhibited in 8505C and FTC 133 cell lines, while the apoptosis rate was markedly increased. Hence, both PI3K/AKT/mTOR and MAPK pathways were thus impacted in this study. This team reported that the dual PI3K/mTOR inhibitor BEZ235 was also tested in combination with sorafenib reporting similar data in thyroid cancer cells [85]. PI3K/AKT and ERK pathways were also the main pathways in a study using the curcumin. However, the low number of thyroid cancer cell lines tested in this study did not lead to solid conclusions about the possible significant beneficial effects in in vivo models, but more investigations should be done considering the good results obtained with curcumin in other types of cancers [84]. A MEK inhibitor, selumetinib in combination with sorafenib was previously tested in the restricted context of MTC. An in vitro synergistic antiproliferative effect was observed, contrary to everolimus for which disappointing results were obtained. Further studies are needed to completely understand feedback loops involved in the latter ineffective combination [83]. Recently, the impact of the sorafenib in combination with the inhibitor of autophagy chloroquine was evaluated. Many studies have reported that sorafenib induces autophagy in cancer cells, as also confirmed in thyroid cancer due to the inhibition of the AKT/mTOR pathway. The process of autophagy is complex in cancer treatment because it may promote or inhibit cell death. Chloroquine, usually used for malaria prophylaxis and the treatment of rheumatoid arthritis, is an autophagy inhibitor, which prevents autophagosomes from fusing with lysosomes, thereby disrupting the process of autophagy. This has been tested in vivo and has shown improved efficacy of sorafenib in treating thyroid cancer xenograft mice. However, due the complexity of the mechanism induced by the combination, it is difficult to clearly established if the two molecules acted in additivity or synergy [86].

Among these combination studies, part of them explored the effects of treatment on ATC. As previously described, ATC is still a challenge in cancer therapy because it is still incurable due to a rapid progression and a high aggressiveness of the tumor cells. The dose of sorafenib use in monotherapy can be reduced by up to 25% if combined with the antidiabetic drug metformin and could thus improve the quality of life of patients as described in the study conducted by Chen et al. This combination inhibited the MAP kinase signaling pathway by reducing ERK phosphorylation [87]. Of course, the MAP kinase signaling pathway is crucial in ATC progression and has also been targeted by the combination of sorafenib and withaferin A, which is a steroidal lactone acting as an anticancer drug. Withaferin A inhibited ERK and AKT phosphorylation, as well as NFκB, thus modulating the cell cycle. In vitro experiments demonstrated promising results in the SW1736 ATC cell line, but future studies examining in vivo models would provide more information about synergistic effects observed in vitro as well as long-term toxicity profile [88]. Additionally, among the categorized histone deacetylase (HDAC) inhibitors, the N-hydroxy-7-(2-naphthylthio) hepatonomide (HNHA) has been associated with sorafenib as a new clinical approach for ATC treatment. The combination therapy has been tested both in vitro and in vivo, showing promising results with an induction of apoptosis by the downregulation of Bcl-2 and an arrest of the cell cycle. The combination of HNHA and sorafenib had a lower IC50 in ATC cells than that of either single agent [89]. Finally, an unexpected molecule was also associated with sorafenib, the antimalaria drug quinacrine. By using molecules already evaluated for the treatment of different human diseases, the authors use a less expensive drug with a well-known safety profile. In vitro and in vivo experiments based on an immunodeficient orthotopic ATC mouse model were performed showing an additive/synergistic tumor cell-killing response with the combined treatment. Inhibition of the expression of the pro-survival genes Mcl-1, pSTAT, and NFκB were observed and an increase of the overall survival of mice was noticed in comparison to mice treated with each compound alone or doxorubicin [82].

As previously approached, lenvatinib alone showed antitumor properties in vivo as a strong antiangiogenic action against human DTC and ATC in xenografts nude mouse model [90]. To potentiate the effects, it has been tested associated with anti-PD-1/PD-L1 therapy in an immunocompetent orthotopic model of ATC. For the last years, the interest in immunotherapies is still growing with many good results obtained in clinical trials, notably combining immunotherapy to targeted therapy as an innovative combinatorial strategy. The addition of the anti-PD-1 to lenvatinib resulted in a dramatic reduction of tumor size and a doubling in mouse survival time after treatment. The effects of this treatment on diverse infiltrating immune cells were evaluated, and an increase of CD8+ within the tumors was observed and enhanced cytotoxicity, as evidenced by higher granzyme B staining and IFNγ production. NK-cells did not significantly change with the combinatorial therapy, while FoxP3+ cells and M2-like TAMs modestly increased despite significant tumor reduction, suggesting a minor role of these immune cells in response to treatment [91]. Finally, the antineoplastic effect of lenvatinib was also evaluated in addition to vandetanib in primary cells from ATC obtained from both biopsy and fine-needle aspiration. The antiproliferative activity of such combination was evaluated in six ATC patients and resulted in a dose-dependent increase of the percentage of apoptosis of the primary tumor cells [44]. The antiangiogenic properties of lenvatinib has been studied with the association of golvatinib, a dual c-Met/VEGFR 2 inhibitor [92]. Analyses suggested that the combination of golvatinib to lenvatinib contributed to the inhibition of the pericyte-mediated vessel stabilization and Tie2-expressing macrophages (TEM) differentiation resulting in a severe perfusion disorder and massive apoptosis. Animal tests revealed a tolerable body weight lost associated with no macroscopic modifications. As increasingly studied in recent years, the modulation of the tumor microenvironment seems to be a crucial element and can be studied by the combination of multitargeting tyrosine kinase inhibitors, which may sensitize cancer to VEGF inhibitors [92]. As for the molecule sorafenib, the lenvatinib has been conjugated to the N-hydroxy-7-(2-naphthylthio) hepatonomide HNHA (an histone deacetylase) and effects of this combination were studied in vitro and in vivo. Results on cells isolated from PTC and ATC of the combinatorial strategy demonstrated an increase of the cell cycle arrest markers but also a suppression of the antiapoptosis markers, the epithelial–mesenchymal transition (EMT), and the FGFR signaling pathway, enhancing the inhibition effect observed on EMT. Recent evidence has demonstrated that induction of the epithelial–mesenchymal transition (EMT) in cancer cells not only induced metastasis, but also serves as a major contributing factor in drug resistance, thus emphasizing the importance of studying this mechanism and its potential inhibition [93]. In comparison to the treatment with sorafenib-HNHA, the association with lenvatinib seems to be even more efficient [89,93]. Finally, animal studies allowed us to show that the combined treatment including lenvatinib induced significant tumor shrinkage [93].

### 2.3. Chemotherapies in Thyroid Cancer Cells

#### 2.3.1. Chemotherapy Used as Single Agent

Doxorubicin, paclitaxel, docetaxel, and cisplatin are common chemotherapies used for the treatment of thyroid cancer. Chemotherapies are the first historical treatments used against cancers and today remain highly unavoidable in many cases. However, their uses are limited due to multiple off-target toxicities, leading not only to a severe impact on the quality of life of patients but also to a reduction of doses or the selection of alternative therapies. One of the heaviest secondary effects observed in the case of doxorubicin treatment is strong cardiotoxicity [94,95]. Cisplatin is sometimes associated with ototoxicity, neurotoxicity, nephrotoxicity, myelosuppression, and nausea/emesis, especially for older patients [96].

#### 2.3.2. Chemotherapy Used in Combination

Avoiding severe secondary effects while optimizing the efficiency of conventional chemotherapy is still a huge challenge in cancer treatment today. That is why its combination with new drugs is now so popular to minimize adverse reactions by reducing the doses of toxic molecules and to play on synergism to achieve the efficiency needed against tumors. Doxorubicin is a quinone containing anthracycline, acting as a DNA intercalator and used for many cancers including ATC, but with the main limitation associated with rapid development of resistance to the drug. To potentiate the efficiency and overcome resistance, different combination strategies have been recently evaluated in the context of thyroid cancer. For example, mitochondrial respiration has essential roles in energy production, as well as in the activation of signaling pathways implicated in the growth and the survival of tumor cells and tumor stem cells. The inhibition of mitochondrial respiratory complex III has been firstly studied using the antiparasitic drug atovaquone. In combination with doxorubicin, atovaquone acted in synergy to inhibit the proliferation of thyroid cancer cells. Moreover, the cardiotoxicity observed with doxorubicin alone seems to be minimized when combined with atovaquone, suggesting that this alternative approach could be promising for aggressive thyroid cancer treatment [97]. Another strategy for ATC has been developed in vitro by combining doxorubicin with cucurbitacin B. Treatment including cucurbitacins alone resulted in inhibition of STAT3. Induction of cytotoxicity through the mediation of Bcl-2 family proteins, surviving an inhibitor of apoptotic protein (IAP) member and ROS, as well as the regulation of JAK2/STAT3 and ERK1/2 were reported in ATC cells, highlighting a synergistic action of the two molecules [98]. Finally, the third approach including a combination of doxorubicin with the XPO1 inhibitor selinexor in the context of ATC brought new hopes in the improvement of the care of these patients. XPO1 is involved in the transport of more than 220 proteins and is overexpressed in many cancers, including ATC, and thus represent an interesting target. An in vivo model reported inhibition of tumor growth in mice treated with doxorubicin plus selinexor, and supported the in vitro results demonstrating a synergistic action between these two molecules. In order to counteract the resistance to the doxorubicin, the combination of SL327 (a MEK1/2 inhibitor) and sunitinib (a multitargeted RTK inhibitor) seems to be a possible option. A side effect could be reduced with this combination in addition to an anticancer efficiency by inducing apoptosis and suppressing viability and migration of ATC cells [99].

Like doxorubicin, paclitaxel is included in current chemotherapy protocols for ATC. However, resistance mechanisms are multiple such as increased activities of ABC transporters as the glycoprotein-p and the BCRP (breast cancer resistance protein), and key mutations in RAS/MAPK/ERK and/or PI3K/AKT/mTOR signaling pathways. Combination with targeted treatments is thus also considered. In particular, the dual mTOR kinase inhibitor AZD2014 has been tested in combination with paclitaxel in the case of ATC in vitro and in vivo. Higher autophagy rates in paclitaxel-resistant cells treated with the combination have been observed and associated with a substantial antiproliferative effect. Inhibition of migration and invasion were considerably improved by cotreatment with AZD2014 (vistusertib) and paclitaxel [100]. These recent results corroborate data from a phase I clinical trial using a combination of AZD2014 and paclitaxel for the treatment of solid tumors [101]. Still concerning ATC studies, paclitaxel has been tested in combination with an HSP90 inhibitor (17-allylamino-17-demethoxygeldanamycin). By binding the N-terminal domain of HSP90 and destabilizing HSP90 client protein, this molecule has shown antitumour activity. However, the relation between the chemotherapy and the HSP90 inhibitor turned out to be an antagonist mechanism of action demonstrating the complexity of the interaction with molecules showing a weak efficiency when used as a single agent [102]. Inhibitors of tyrosine kinase receptors as VEGF receptors have been evaluated with paclitaxel and are described in the next paragraph.

Studies combining chemotherapies with other categories of molecules targeting MAP kinase and PI3K/mTOR pathways are still ongoing. By combining the docetaxel (acting by disturbing microtubule disassembly and used to treat metastatic cancers like prostate cancer) and the flavone baicalein (inhibiting ERK and AKT/mTOR), Park et al. observed a strong downregulation of the expression of apoptotic and angiogenic proteins in ATC cells [103]. Used with the histone deacetylase inhibitor, suberoylanilide hydroxamic acid (SAHA), docetaxel represents a novel approach by acting on the NF-kB pathway. In vivo assays have demonstrated a reduction of tumor growth, but only in one cell line, diminishing the enthusiasm for this approach. Further research is needed to broadly apply this kind of treatment, combining conventional chemotherapy and novel targeted therapy to patients with advanced thyroid cancer [104].

Altogether, preclinical data demonstrated that a combination of therapies, including an agent of chemotherapy, are, in some cases, very promising new therapeutic strategies and deserve to be tested in clinical studies, especially for ATC. Nevertheless, in other cases, disappointing results have been obtained which support the necessity to continue investigations to better and deeply understand the mechanisms of resistance through the interconnection of cellular pathways.

#### 2.3.3. Chemotherapies and VEGF Inhibitors

Chemotherapies and tyrosine kinase inhibitors, such as VEGF receptor inhibitors, have shown limited effects when used alone. As discussed above, chemotherapies need to be associated with other categories of molecules to improve the efficacy of the treatment and reduce the side effects.

MTC form is more rarely studied in the context of combination therapies, but Lopergolo et al. investigate the effect of the chemotherapy cisplatin completed with the multikinase inhibitor sunitinib. Since the RET receptor tyrosine kinase is highly implicated in the pathogenesis of both inherited and sporadic MTCs, sunitinib, which targets both angiogenesis-related receptors and RET, combined with cisplatin, was viewed as a promising therapeutic approach. In vitro and in vivo experiments demonstrated a huge activation of the extrinsic apoptotic pathway and alterations of the lysosomal functions [105].

Sunitinib was approved for second-line treatment of advanced renal cell carcinomas and advanced gastrointestinal stromal tumors after resistance to imatinib therapy. It was evaluated in ATC combined with the chemotherapeutic agent irinotecan used as a standard molecule for colorectal cancer. This study highlighted a synergistic inhibitor effect of these two molecules on ATC cell proliferation explained by a significant increase of SN-38 (the active metabolite of irinotecan) intracellular levels mediated by sunitinib. Moreover, a significant in vivo antitumor effect has been observed in ATC xenografts. As we already know, thyroid cancers are heavily infiltrated with macrophages and 95% of anaplastic thyroid cancer cases showed high tumor associated macrophages (TAMs) infiltration, which correlated with a poor survival rate. Macrophages are dependent of CSF-1 for differentiation and survival and, at high density, studies demonstrated their influence on the overall gene expression profile in ATC. In this study, they found a significant reduction of CSF-1 gene expression and protein levels due to the simultaneous schedules with sunitinib and SN-38 in ATC cells. Data showed that the decline of CSF-1 levels in ATC could decrease the macrophages stimulation induced by this myelopoietic growth factor in vivo [106].

As previously described, paclitaxel was also tested in combination with many molecules. In ATC, the two molecules lenvatinib (VEGF inhibitor) and pazopanib (VEGF, PDGF, and c-kit inhibitor) have been associated with paclitaxel. Both molecules were approved for advanced/metastatic renal cell carcinomas, and pazopanib was added for advanced soft tissue sarcomas. These molecules potentiated the effect of paclitaxel in vitro and in vivo. In the case of pazopanib plus paclitaxel, the effects were associated with the inhibition of aurora A and the alteration of mitosis [107,108] and were strong enough to propose a clinical study. It is important to note that Aurora A has been found to be overexpressed in 41% of ATC patient samples [109]. The second study using paclitaxel investigated the effect in vitro and in vivo in addition to lenvatinib. A synergistic inhibition of colony formation and tumor growth in nude mice was observed in addition to a G2/M phase cell cycle arrest and cell apoptosis [108].

## 3. Clinical Studies Combining Therapies in Thyroid Cancer Cells

The association of different drugs in clinical studies brings hope to patients for whom therapeutic strategies have been ineffective thus far. Given that promising results are obtained in preclinical studies or tested in the context of another form of cancer, many clinical studies testing a combination of treatments have been conducted or are still ongoing. Chemotherapies seem still today to remain unavoidable and are often combined with other categories of molecules. The categories of targeted therapy are frequently tested in combination with one of the gold standard chemotherapies associated to the diagnosed form of thyroid cancer. The significant results for these molecules in preclinical studies have established rationales for clinical evaluations. Ljubas et al. recently published a systematic review about phase II targeted therapy clinical trials specifically in ATC [110]. In the third part of our review, we reported the results from the treatment combination examined in clinical studies for all types of thyroid cancers, as well as the results of the ongoing studies presenting the perspectives of treatment for resistant forms of thyroid cancer to conventional therapies (Table 2).

### 3.1. Closed Clinical Studies

Currently, eleven closed clinical studies have evaluated the efficacy of combinatorial therapies in thyroid cancer as summarized in Table 3. Many of them used chemotherapies as the main agents for treatments, while others combined targeted therapies or propose to target oncogenic fusions.

#### 3.1.1. Combination with Chemotherapy

Catalano et al. investigated a phase II/III trial combining the valproic acid (VPA), a histone deacetylase inhibitor, traditionally used for epilepsy treatment, with paclitaxel for ATC between 2009 and 2012 across five centers. The median survival for patients treated by the combination recorded 122 days. Patients treated with the combination did not experience more side effects than those treated by paclitaxel alone. However, the trial demonstrated that the combination of VPA and paclitaxel was not superior to paclitaxel alone to induce a response in cancer patients [111].

A phase I clinical trial targeting the anaplastic form of thyroid cancer and implicating the paclitaxel with the efatutazone (oral PPAR-γ agonist) reported that the combination was safe and well-tolerated. Primarily based on the good results obtained on cell lines and with the xenograft model, this clinical study provided encouraging pilot data in support of further trials combining paclitaxel and efatutazone in ATC. More effort should be made to extend these encouraging results and to develop a definitive randomized trial assessing the incremental effects of the addition of efatutazone to paclitaxel monotherapy in patients with advanced ATC [112].

A tri-therapy was also conducted, adding paclitaxel to combretastatin and carboplatin in ATC. The purpose of the study (NCT00507429) was to determine the safety and efficacy of combretastatin combined with paclitaxel and carboplatin in the treatment of ATC. A total of 80 patients were included, with 55 of them in the tri-therapy group. Due to disease progression (32%), heavy adverse events (32%), patient withdrawal (8%), physician’s decision (11%), or for other reasons (16%), 67% of patients did not complete the study. These results indicated that the toxicity or the ineffectiveness depended on the case and lessened the hope of the very promising results obtained in a nude mouse xenograft model [113].

In two other studies, the efficiency of capecitabine was evaluated in association with dacarbazine and the multikinase inhibitor imatinib. Preclinical studies suggested a potential activity of imatinib in endocrine cancers, including MTC. Associated with the cytotoxic chemotherapies capecitabine and dacarbazine, the safety profile of this new treatment was evaluated on 20 patients, 7 of whom had an MTC. The most frequent toxicities were oedema and fatigue, with dose-limiting fatigue and dyspnea. In perspective of phase II, the recommended plan of treatment should be dacarbazine 250 mg/m^2^ daily on day 1–3, capecitabine 500 mg/m^2^ twice daily on days 1–14, and imatinib 300 mg daily on days 1–21 of a 21 day cycle. However, the authors pointed out the fact that multiple studies have investigated the in vitro and in vivo activity of imatinib-based regimens in MTC [114,115]. The effects demonstrated RET inhibition and death of the oncogene-addicted MTC cells but only at serum concentrations that could not be achieved in patients with tolerable doses of imatinib, suggesting weak effectiveness of such combination in MTC [116].

Another closed trial including capecitabine was a phase I clinical study exploring its association with irofulven, a semisynthetic derivative of illudin able to inhibit DNA synthesis. The objectives of this study were to determine the maximum tolerated dose (MTD), the recommended dose, the dose-limiting toxicities (DLT), the safety and pharmacokinetics of irofulven combined with capecitabine in advanced solid tumor patients, including four patients with a thyroid tumor. Results demonstrated that irofulven with capecitabine was adequately tolerated and showed evidence of an antitumour activity. Two partial responses were noted in thyroid carcinoma. Among prospects for a future phase II, the doses, which would be used are 0.4 mg/kg for irofulven and 2000 mg/m^2^/day for capecitabine [101].

Finally, the combination of cisplatin plus crolibulin (a tubulin polymerization inhibitor) was evaluated on 26 patients with ATC. Patients received escalating doses (75–100 mg/m^2^ cisplatin and 8–20 mg/m^2^ crolibulin) depending on the study group. Adverse events were observed as anemia, increased creatinine level, nausea, or hypertension. Unfortunately, phase II was not completed because of the lack of recruitment [117].

#### 3.1.2. Combination with Targeted Therapies

As previously introduced, targeted therapies are now part of the explored therapies in thyroid cancer. Six studies evaluated the combination of these molecules to define better treatment strategies.

The molecule of pazopanib previously described in the preclinical part of our review was implicated in a phase I trial in combination with escalating doses of 131I in patients with well-differentiated thyroid carcinomas and borderline refractory to radioiodine. Hence, this trial was conducted on six patients to evaluate the ability of pazopanib to overcome 131I therapeutic resistance. However, this study reported no convincing impact of pazopanib on iodine uptake, in scans performed pre- and post-therapy, compared to scans from patients with 131I treatment without pazopanib. Despite a suggestion of therapeutic efficacy, the association of pazopanib and 131I resulted in increased toxicity [118].

A study exploring the effects of the combination of vandetanib and bortezomib ended, unfortunately, with disappointing results. The authors succeeded to establish the recommended phase II dose for the combination (vandetanib 300 mg orally daily and bortezomib 1.3 mg/m^2^ intravenously on days 1, 4, 8, and 11), however the efficacy of this treatment in the 17 patients with MTC was not sufficient to continue and propose it as a therapeutic option in MTC [119].

The next closed studies that we reviewed all result from phase II trials. A recent trial evaluated the effect of pasireotide LAR (a somatostatin analogue) alone and in combination with the mTOR inhibitor everolimus in patients with MTC. Seven patients were treated with pasireotide LAR plus everolimus, and the median PFS was 9 months (95% CI: 0–22). Tumor progression was detected for five of them, while one patient discontinued the treatment for the occurrence of pulmonary embolism. An objective response was observed in only one patient after 6 months of combinatorial treatment. Although the association of pasireotide LAR with everolimus could be viewed as an alternative treatment in order to delay tumor progression in a subgroup of patients with postoperative persistent MTC and with small and not progressive lesions, the real benefit of this combination in MTC remains to be established in random trials on a larger series of patients [120].

Furthermore, Sherman et al. evaluated the potential efficacy of the association of the multikinase inhibitor sorafenib and the temsirolimus (a mTOR inhibitor) in patients with radioactive iodine refractory thyroid cancer. Sorafenib alone is already approved for treatment by the Food and Drug Administration (FDA) in the case of advanced renal carcinoma (2005), unresectable hepatocellular carcinoma (2007) and locally recurrent or metastatic, progressive differentiated thyroid carcinoma (DTC) refractory to radioactive iodine treatment (2013) [39]. The inhibitory effects of sorafenib were evaluated in the international multicentric phase III DECISION study. Used as a single agent, this study did not demonstrate any pieces of evidence of improvement of patient overall survival. Moreover, patients often acquired resistance after 1 or 2 years of sorafenib administration [69]. Finally, toxic side effects such as hand–foot syndrome, diarrhea, fatigue, rash, and weight loss affected approximately 80% of sorafenib-treated patients, rendering this molecule not useable alone for cancer treatment. Then combination has been proposed through the administration of oral sorafenib (200 mg twice daily) and intravenous temsirolimus (25 mg weekly). The best results obtained were a partial response in 8 out of 36 patients (22%), stable disease in 21 (58%), and progressive disease in 1 (3%). The sorafenib and temsirolimus combination seems to be more effective in patients with radioactive iodine-refractory thyroid cancer, than sorafenib alone. It is also important to note that two of these patients presented an ATC and one of them experienced a major response to sorafenib and temsirolimus. It could therefore be interesting to extend the targeted population to patients with ATC in further investigations [121].

In addition, the kinase inhibitors sorafenib, vandetanib, cabozantinib, and lenvatinib are molecules approved by the FDA and the European Medical Agency (EMA) for the treatment of MTC (vandetanib and cabozantinib) and advanced RAI-R (radioactive iodine therapy refractory) and poorly differentiated thyroid cancer (PDTC; sorafenib and lenvatinib) [39]. Globally, the main critic for vandetanib, cabozantinib, and sorafenib is that the overall survival for the patients is not improved based on the ZETA, EXAM, and DECISION clinical studies. Concerning lenvatinib, an increase of overall survival has been observed in the SELECT study, but selectively for the subgroup of patients >65 years old [39,70].

The final closed study detailed in this review investigated the response to the association of the two targeted therapies dabrafenib (mutant BRAF inhibitor) and trametinib (MEK inhibitor) targeting the MAPK pathway in patients with locally advanced or metastatic V600EBRAF ATC. Among the enrolled patients, many of them had prior treatments, including surgery (88%), external beam radiotherapy (81%), and chemotherapy (38%). A robust clinical activity was demonstrated by such combination. Dabrafenib plus trametinib treatment resulted in a confirmed objective response rate of 69%. The responses typically occurred early in the treatment course. The authors noticed the disappearance of multiple pulmonary metastases in complete responders within the first 8 weeks of therapy. Moreover, the responses were also durable as 90% of patients were always responders at 12 months of treatment, reporting a promising new combination for patients with mutant BRAF ATC [122]. Hence, the clinical study combining dabrafenib and trametinib bring new exciting perspectives of treatment for patients to whom no effective therapy is available until now.

### 3.2. Ongoing Clinical Studies

The innovative combinatorial treatments are numerous as documented by the number of studies recruiting patients (Table 2). Chemotherapies, targeted therapies, and immunotherapy are the categories of molecules mainly used in the combinatorial therapies. The future will allow us to see if new combinations of molecules can obtain partial or complete responses in thyroid cancer patients who currently have no effective therapy.

## 4. Targeting Gene Rearrangements in Thyroid Cancer

Gene rearrangements result in the aberrant activity of tyrosine kinases and are identified as driving alterations in a few rare types of cancer. Mechanisms of resistance to targeted therapies are complex and are still today no deeply understood. Gene fusions in thyroid cancer (e.g., RET-PTC 1-12 and NTRK) could be involved in such mechanisms because of their actions on the activation of the mitogen-activated protein kinase (MAPK) pathway. Gene fusions are more common in PTC compared to other histological subtypes. Understanding how genetic alterations contribute to the disease process is crucial for the emergence of novel prognostic and therapeutic strategies [123].

Among them, NTRK gene fusions are well-established as oncogenic events in specific cancer, especially in papillary thyroid carcinoma [124]. Actually, the tropomyosin receptor kinase (TRK) family contains three members, TRKA, TRKB, and TRKC, which are encoded by the genes NTRK1, NTRK2, and NTRK3, respectively [125].

For patients carrying such alterations, selective small-molecule inhibitors of the TRK kinases, such as larotrectinib, emerges as important novel therapeutic options [126]. Indeed, larotrectinib has been recently approved by the FDA for NTRK-fusion thyroid cancer [127]. To date, multiple clinical trials testing TRK-inhibiting compounds in various cancers are ongoing, and NTRK inhibitors, alone or combined with immune checkpoint inhibitors, may be a new therapeutic option for NTRK-fusion cancer patients [128].

Additionally, the chromosomal rearrangement of the ALK gene to various partners leads to a constitutive activation of ALK tyrosine kinase and represents a carcinogenic mechanism found in various cancer type including non-small-cell lung cancer and thyroid cancer [129]. In thyroid cancers, ALK fusions, most frequently the STRN-ALK fusion, are detected in PTC and, with higher frequency, in PDTC and ATC. The crucial role of the *STRN-ALK* fusion in the development of dedifferentiated thyroid cancer have been highlighted by the development of a transgenic mouse model with phenotypic features recapitulating a poor differentiated human tumor [129].

In the context of RAI-R disease, the study of the gene fusions seems to be promising. Indeed, the identification of gene fusion transcripts leading to the activation of signaling pathways impacting the restoration of iodine transport process should bring additional targets to fight the resistance occurring in RAI-PTC and RAI-HCC [123]. Therefore, the ALK translocations in thyroid carcinoma represents an interesting therapeutic approach especially because such translocations are over-represented in clinically aggressive thyroid carcinomas and could be inactivated in combination with targeted therapies or chemotherapies to block the resistance mechanisms [130].

## 5. Conclusions

Due to the number of publications addressing treatment combinations for the various forms of thyroid cancer, it is clear that this strategy is now unavoidable. Many preclinical studies provide encouraging results for aggressive forms of cancer, such as anaplastic cancers or radio-iodine refractory papillary thyroid cancers. Although the results are more questionable in clinical studies, the combination of therapies, including targeted therapies and conventional chemotherapies, currently provide a treatment opportunity for patients whose tumors are resistant to traditional monotherapies. However, it remains essential to fully understand the different mechanisms of resistance developed by tumors to better break them down in the future using the best fit targeted combinatorial treatments for each patient.

## Figures and Tables

**Figure 1 cells-09-00830-f001:**
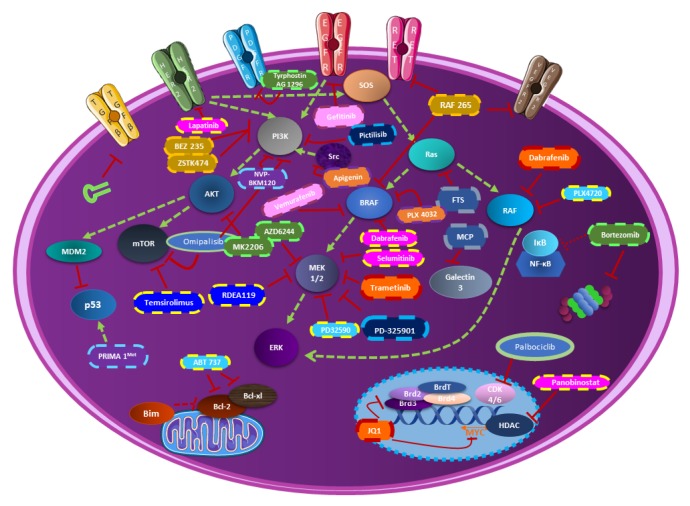
The combination of PI3K-MAPK-mTOR inhibitors in thyroid cancer in preclinical studies. This figure resumes all combinatorial therapies including PI3K-MAPK-mTOR inhibitors in thyroid cancer. Each molecule represented with a specific color legend is associated with the molecules having the same color legend.

**Figure 2 cells-09-00830-f002:**
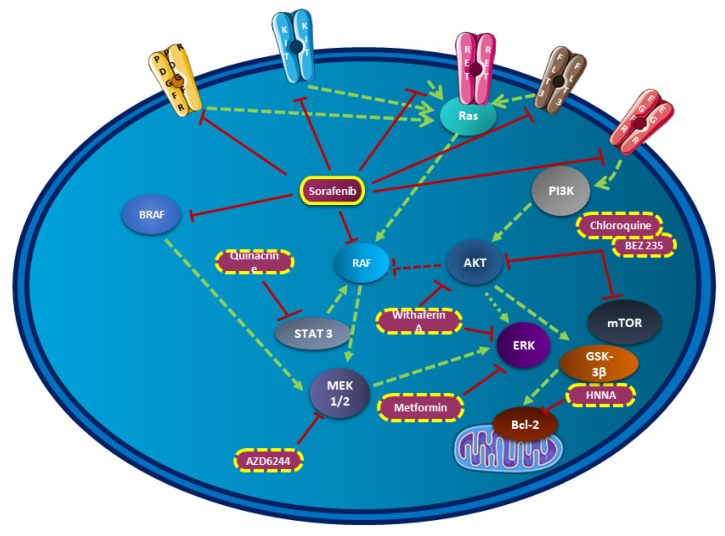
Sorafenib combination in thyroid cancer in preclinical studies. This figure resumes all molecules tested in preclinical studies associated with sorafenib in thyroid cancer and their implications in crucial signaling pathways.

**Table 1 cells-09-00830-t001:** Summary of preclinical studies using combinatorial therapies.

Thyroid Cancer Type	Cell Line	Species	Mutation	Molecular Target	Combination of Therapies	References
Anaplastic form	SW1736	Human	Heterozygous BRAFV600E, Heterozygous for TERT c.228C >T (-124C >T), Homozygous for TP53 p.Gln192Ter, Heterozygous for TSHR p.Ile486Phe	BRAF^V600E^ inhibitor/proteasome inhibitor	Vemurafenib + Bortezomib	Tsumagari et al., 2018
				MEK1/2 inhibitor/mTOR inhibitor	RDEA119 + temsirolimus	Liu et al., 2010
				Raf inhibitor/immunotherapy	PLX4720 + anti PDL-1	Brauner et al., 2015
				Withanolide (potent of heat-shock protein inhibition)/VEGF inhibitor	Withaferin A + Sorafenib	Cohen et al., 2012
				Selective inhibitor of nuclear export/chemotherapy	Selinexor + doxorubicin	Garg et al., 2017
				mTOR inhibitor/chemotherapy	AZD2014 + paclitaxel	Milošević et al., 2018
				Histone deacetylase inhibitor/chemotherapy	Suberoylanilide hydroxamic acid (SAHA) + docetaxel	Pozdeyev et al., 2015
	KAT 18	Human	Heterozygous for MRE11A p.Leu57Ter, Heterozygous for TERT c.228C >T (−124C >T), Homozygous for TP53 p.Gly199Val (c.596G >T)	BRAF^V600E^ inhibitor/proteasome inhibitor	Vemurafenib + Bortezomib	Tsumagari et al., 2018
				MEK1/2 inhibitor/mTOR inhibitor	RDEA119 + temsirolimus	Liu et al., 2010
	KAT 4	Human	Heterozygous for APC p.Glu853Ter (c.2557G >T) and p.Thr1556fs*3 (c.4666_4667insA), Heterozygous for BRAF p.Val600Glu (c.1799T >A), Heterozygous for PIK3CA p.Pro449Thr (c.1345C >A), Homozygous for SMAD4 p.Gln311Ter (c.931C >T), Homozygous for TP53 p.Arg273His (c.818G >A)	AKT inhibitor/platelet-derived growth factor receptor inhibitor	MK-2206 + tyrphotsin AG 1296	Che et al., 2014
				MEK1/2 inhibitor/VEGF inhibitor	SL327 + sunitinib	Wang et al., 2017
	8505c	Human	Homozygous for BRAF p.Val600Glu, Homozygous for NF2 p.Glu129Ter, Heterozygous for TERT c.250C >T, Homozygous for TP53 p.Arg248Gly	Bcl2 family inhibitor/MAPK inhibitors	ABT-737 + PLX4720 + PD32590	Gunda et al., 2017
				SRC inhibitor/Raf inhibitor	Dasatinib + PLX4720	Vanden Borre et al., 2014
				Flavonoid derivative/BRAFV600E inhibitor/PI3K inhibitor	Apigenin + vemurafenib + wortmannin	Kim et al., 2013
				Raf inhibitor/immunotherapy	PLX4720 + anti PDL-1	Brauner et al., 2015
				Raf inhibitor/PI3K inhibitor	RAF265 + Dactolisib (BEZ-235)	Jin et al., 2011
				Growth factor inhibition/AKT inhibitor	shRNA TGF-β1 + MK-2206	Li et al., 2016
				BRAF^V600E^ inhibitor/c-Met inhibitor	Vemurafenib + PHA665752	Byeon et al., 2017
				Mek inhibitor/PI3K inhibitor	PD-325901 + GDC-0941	ElMokh et al., 2017
				VEGF inhibitor/PI3K inhibitor/AKT inhibition	Sorafenib + Dactolisib (BEZ235)/small interfering RNA (siRNA) directed against AKT	Yi H et al.,2017
				VEGF inhibitor/antimalaria drug	Sorafenib + Quinacrine	Abdulghani et al., 2016
				VEGF inhibitor/antimalaria drug	Sorafenib + chloroquine	Yi H et al.,2018
				VEGF inhibitor/Histone deacetylase inhibitor	Sorafenib + N-hydroxy-7-(2-naphthylthio) hepatonomide (HNHA)	Cheong Park et al., 2017
				anti parasitic drug/chemotherapy	Atovaquone + Doxorubicin	Zhuo Lv et al., 2018
				Oxidized tetracyclic triterpenoids (inhibitor JAK/STAT)/chemotherapy	Cucurbitacin B + Doxorubicin	Hyoung Kim et al., 2017
				mTOR inhibitor/chemotherapy	AZD2014 + paclitaxel	Milošević et al., 2018
				Antibiotic/Chemotherapy	Tanespimycine (17-allylamino-17-demethoxygeldanamycin) + paclitaxel	Kim et al., 2014
				Bioactive flavone/chemotherapy	Baicalein + docetaxel	Ho Park et al., 2018
				Histone deacetylase inhibitor/chemotherapy	Suberoylanilide hydroxamic acid (SAHA) + docetaxel	Pozdeyev et al., 2015
				VEGF inhibitor/chemotherapy	Lenvatinib + Paclitaxel	Jing et al., 2017
				BRAF inhibitor/TKI	Vemurafenib + Ponatinib	Ghosh et al., 2020
				MEK inhibitor/dietary supplement	U0126 + sodium selenite	Kim et al., 2020
	HTh-7	Human	NRAS p.Gln61Arg (c.182A >G), KMT2D p.Gln4118Ter, Homozygous for TERT c.250C >T, TP53 p.Gly245Ser	Bcl2 family inhibitor/MAPK inhibitors	ABT-737 + PLX4720 + PD32590	Gunda et al., 2017
				Raf inhibitor/immunotherapy	PLX4720 + anti PDL-1	Brauner et al., 2015
				Selective inhibitor of nuclear export/chemotherapy	Selinexor + doxorubicin	Garg et al., 2017
	TBP-3743	Murine	BrafV600E/WT; p53/	SRC inhibitor/Raf inhibitor	Dasatinib + PLX4720	Vanden Borre et al., 2014
				Immunotherapy/VEGF inhibitor	Anti PDL1 + lenvatinib	Gunda et al., 2019
	TBPt-3403	Murine	BrafV600E/WT; Pten−/−	SRC inhibitor/Raf inhibitor	Dasatinib + PLX4720	Vanden Borre et al., 2014
	TBPt-3610R	Murine	BrafV600E/WT; Pten−/−	SRC inhibitor/Raf inhibitor	Dasatinib + PLX4720	Vanden Borre et al., 2014
	FRO	Human	Heterozygous BRAFV600E mutation	Flavonoid derivative/BRAFV600E inhibitor/PI3K inhibitor	Apigenin + vemurafenib + wortmannin	Kim et al., 2013
				VEGF inhibitor/chemotherapy	Pazopanib + Paclitaxel	Isham et al., 2013
	ACT-1	Human	Heterozygous for NRAS p.Gln61Lys, Heterozygous for TERT c.250C >T, Homozygous for TP53 p.Cys242Ser	MAPK inhibitors	Trametinib + dabrafenib	Kurata et al., 2016
	OCUT-1	Human	Heterozygous for BRAF p.Val600Glu (c.1799T >A), Heterozygous for TERT c.228C >T (-124C >T), Homozygous for TERT c.250C >T	MEK1/2 inhibitor/mTOR inhibitor	RDEA119 + temsirolimus	Liu et al., 2010
				AKT inhibitor/MEK inhibitor	MK-2206 + selumetinib	Liu et al., 2012
	OCUT-2	Human	Heterozygous for BRAF p.Val600Glu, Homozygous for TERT c.250C >T	MAPK inhibitors	Trametinib + dabrafenib	Kurata et al., 2016
	OCUT-4	Human	BRAF and PI3KCA mutations	MAPK inhibitors	Trametinib + dabrafenib	Kurata et al., 2016
	OCUT-6	Human	Wildtype BRAF and NRAS mutations	MAPK inhibitors	Trametinib + dabrafenib	Kurata et al., 2016
	THJ-11T	Human	KRAS p.Gly12Val (c.35G >T), Heterozygous for TERT c.228C >T	MEK inhibitor/BET inhibitor	Trametinib + JQ1	Zhu et al., 2018
				VEGF inhibitor/chemotherapy	Pazopanib + Paclitaxel	Isham et al., 2013
	THJ-16T	Human	MKRN1-BRAF in-frame gene fusion, Heterozygous for PIK3CA p.Glu545Lys, Homozygous for EP300 p.Ser799Phefs*5, Heterozygous for RET p.Glu90Lys, Heterozygous for TERT c.228C >T, Homozygous for TP53 p.Arg273His	MEK inhibitor/BET inhibitor	Trametinib + JQ1	Zhu et al., 2018
				PI3K inhibitor/p53 reactivator	NVP-BKM120 + PRIMA-1Met	Li et al., 2018
				VEGF inhibitor/antimalaria drug	Sorafenib + Quinacrine	Abdulghani et al., 2016
				Histone deacetylase inhibitor + chemotherapy	Suberoylanilide hydroxamic acid (SAHA) + docetaxel	Pozdeyev et al., 2015
				VEGF inhibitor/chemotherapy	Pazopanib + Paclitaxel	Isham et al., 2013
				BRAF inhibitor + TKI	Vemurafenib + Ponatinib	Ghosh et al., 2020
	THJ-21T	Human	Homozygous for BRAF p.Val600Glu (c.1799T >A), Heterozygous for TERT c.228C >T (-124C >T); in promoter, Homozygous for TP53 p.Arg280Thr (c.839G >C)	PI3K inhibitor/p53 reactivator	NVP-BKM120 + PRIMA-1Met	Li et al., 2018
				VEGF inhibitor/antimalaria drug	Sorafenib + Quinacrine	Abdulghani et al., 2016
				VEGF inhibitor/chemotherapy	Pazopanib + Paclitaxel	Isham et al., 2013
	THJ-29T	Human	FGFR2-OGDH in-frame gene fusion, Homozygous for CDKN2A p.Gln70Serfs*102 (c.207delG) (G55fs), Heterozygous for HDAC10 p.His134Thrfs*19 (c.399delG), Heterozygous for TERT c.250C >T (-146C >T); in promoter, Homozygous for TP53 p.Gln104Ter (c.310C >T)	PI3K inhibitor/p53 reactivator	NVP-BKM120 + PRIMA-1Met	Li et al., 2018
				VEGF inhibitor/antimalaria drug	Sorafenib + Quinacrine	Abdulghani et al., 2016
				VEGF inhibitor/chemotherapy	Pazopanib + Paclitaxel	Isham et al., 2013
	Hth-74	Human	Homozygous for NF1 p.Leu732fs (c.2195_2202delTGCCCAAC), Homozygous for TERT c.228C >T (-124C >T)	PI3K inhibitor/p53 reactivator	NVP-BKM120 + PRIMA-1Met	Li et al., 2018
				VEGF inhibitor/anti diabetic drug	Sorafenib + metformin	Chen et al., 2015
				Selective inhibitor of nuclear export/chemotherapy	Selinexor + doxorubicin	Garg et al., 2017
	ARO	Human	Heterozygous for APC p.Glu853Ter (c.2557G >T), Thr1556fs*3 (c.4666_4667insA), Heterozygous for BRAF p.Val600Glu (c.1799T >A), Heterozygous for PIK3CA p.Pro449Thr (c.1345C >A), Homozygous for SMAD4 p.Gln311Ter (c.931C >T), Homozygous for TP53 p.Arg273His (c.818G >A)	Ras inhibitor/galectin 3 inhibitor	Transfarnesylthiosalicylic acid (FTS) + Modified citrus pectin (MCP)	Menachem et al., 2015
	HTOR	Human	X-Normal thyroid cells	Raf inhibitor/immunotherapy	PLX4720 + anti PDL-1	Brauner et al., 2015
				PI3K inhibitor/p53 reactivator	NVP-BKM120 + PRIMA-1Met	Li et al., 2018
	CAL-62	Human	Heterozygous for CREBBP p.Glu1541Ter (c.4621G >T), Homozygous for EP300 p.Asp1485fs (c.4454delA), Homozygous for KRAS p.Gly12Arg (c.34G >C), Homozygous for NF2 p.Glu215Ter (c.643G >T), Homozygous for TP53 p.Ala161Asp (c.482C >A)	Raf inhibitor/PI3K inhibitor	RAF265 + Dactolisib (BEZ-235)	Jin et al., 2011
				AKT inhibitor/platelet-derived growth factor receptor inhibitor	MK-2206 + tyrphotsin AG 1296	Che et al., 2014
				Oxidized tetracyclic triterpenoids (inhibitor JAK/STAT)/chemotherapy	Cucurbitacin B + doxorubicin	Hyoung Kim et al., 2017
				Selective inhibitor of nuclear export/chemotherapy	Selinexor + doxorubicin	Garg et al., 2017
				MEK inhibitor/VEGF inhibitor	SL327 + sunitinib	Wang et al., 2017
				Antibiotic/Chemotherapy	Tanespimycine (17-allylamino-17-demethoxygeldanamycin) + paclitaxel	Kim et al., 2014
	C643	Human	HRAS p.Gly13Arg (c.37G >C), Heterozygous for PTEN p.Phe341Leu (c.1023T >G), Heterozygous for TERT c.228C >T (-124C >T), Homozygous for TP53 p.Arg248Gln (c.743G >A), VTCN1 p.Tyr215Ter (c.645C >G)	Raf inhibitor/PI3K inhibitor	RAF265 + BEZ-235	Jin et al., 2011
				VEGF inhibitor/chemotherapy	Lenvatinib + Paclitaxel	Jing et al., 2017
				BRAF inhibitor/TKI	Vemurafenib + Ponatinib	Ghosh et al., 2020
	SNU-80	Human	BRAF p.Gly469Arg (c.1405G >C), Heterozygous for TP53 p.Pro278Ala (c.832C >G)	Histone deacetylase inhibitor/VEGF inhibitor	N-hydroxy-7-(2-naphthylthio) hepatonomide (HNHA) + Sorafenib	Cheong Park et al., 2017
	GSA1	Human	X	Histone deacetylase inhibitor/VEGF inhibitor	N-hydroxy-7-(2-naphthylthio) hepatonomide (HNHA) + Sorafenib	Cheong Park et al., 2017
	T238	Human	BRAF p.Val600Glu (c.1799T >A),Homozygous for CDKN2A p.Leu63Arg (c.188T >G),Heterozygous for PIK3CA p.Glu542Lys (c.1624G >A),Heterozygous for TERT c.228C >T (-124C >T),Homozygous for TP53 p.Ser183Ter (c.548C >G)	Selective inhibitor of nuclear export/chemotherapy	Selinexor + doxorubicin	Garg et al., 2017
	Hth-83	Human	Homozygous for AR p.Gly456_Gly457insGly (c.1368_1369insGGA), Heterozygous for HRAS p.Gln61Arg (c.182A >G),Heterozygous for TERT c.228C >T (-124C >T),Heterozygous for TP53 p.Pro153Alafs*28	Selective inhibitor of nuclear export/chemotherapy	Selinexor + doxorubicin	Garg et al., 2017
	8305C	Human	Heterozygous for ATM p.Gln2800Ter (c.8398C >T),Heterozygous for BRAF p.Val600Glu (c.1799T >A),Heterozygous for NRAS p.Phe90fs (c.270delT),Homozygous for TP53 p.Arg273Cys (c.817C >T),Heterozygous for TERT c.250C >T (-146C >T)	Chemotherapy/VEGF inhibitor	Irinotecan + sunitinib	Di Desidero et al., 2017
				VEGF inhibitor/chemotherapy	Lenvatinib + Paclitaxel	Jing et al., 2017
	FB3	Human	X	Chemotherapy/VEGF inhibitor	Irinotecan + sunitinib	Di Desidero et al., 2017
	KTC-1	Human	Heterozygous for BRAF p.Val600Glu, Heterozygous for RAC1 p.Asp63Val, Heterozygous for TERT c.250C >T	VEGF inhibitor/chemotherapy	Pazopanib + Paclitaxel	Isham et al., 2013
	KTC-2	Human	Heterozygous for BRAF p.Val600Glu (c.1799T >A),Heterozygous for KMT2D p.Glu490Ter (c.1468G >T),Heterozygous for TERT c.228C >T (-124C >T)	VEGF inhibitor/chemotherapy	Pazopanib + Paclitaxel	Isham et al., 2013
	KTC-3	Human	X	VEGF inhibitor/chemotherapy	Pazopanib + Paclitaxel	Isham et al., 2013
Papillary form	BCPAP	Human	Homozygous for BRAF p.Val600Glu, Heterozygous for TERT c.228C >T, Homozygous for TP53 p.Asp259Tyr	MEK inhibitor/VEGF inhibitor	Dabrafenib/selumetinib + lapatinib	Cheng et al.,2017
				Bcl2 family inhibitor/MAPK inhibitors	ABT-737 + PLX4720 + PD32590	Gunda et al., 2017
				SRC inhibitor/Raf inhibitor	Dasatinib + PLX4720	Vanden Borre et al., 2014
				MEK1/2 inhibitor/mTOR inhibitor	RDEA119 + temsirolimus	Liu et al., 2010
				Raf inhibitor/immunotherapy	PLX4720 + anti PDL-1	Brauner et al., 2015
				Raf inhibitor/PI3K inhibitor	RAF265 + Dactolisib (BEZ-235)	Jin et al., 2011
				PI3K inhibitor/p53 reactivator	NVP-BKM120 + PRIMA-1Met	Li et al., 2018
				BRAFV600E inhibitor/c-met inhibitor	PLX4032 + PHA665752	Byeon et al., 2017
				Natural withanolide/VEGF inhibitor	Withaferin A + Sorafenib	Cohen et al., 2012
				Histone deacetylase inhibitor + chemotherapy	Suberoylanilide hydroxamic acid (SAHA) + docetaxel	Pozdeyev et al., 2015
				BRAF inhibitor/TKI	Vemurafenib + Ponatinib	Ghosh et al., 2020
				BRAF inhibitor/EZH2 inhibitor	Selumetinib/dabrafenib + tazemetostat	Fu et al., 2020
	K1	Human	Heterozygous for BRAF p.Val600Glu, Heterozygous for PIK3CA p.Glu542Lys, Heterozygous for TERT c.228C >T	MEK inhibitor/VEGF inhibitor	Dabrafenib/selumetinib + lapatinib	Cheng et al.,2017
				AKT inhibitor/MEK inhibitor	MK-2206 + selumetinib	Liu et al., 2012
				BRAF inhibitor/EZH2 inhibitor	Selumetinib/dabrafenib + tazemetostat	Fu et al., 2020
	BHP 2-7	Human	CCDC6-RET (RET/PTC1) gene fusion, Homozygous for CDKN2A p.Ala68fs, Heterozygous for STAG2 p.Gln1089Ter, Heterozygous for TERT c.228C >T	MEK inhibitor/VEGF inhibitor	Dabrafenib/selumetinib + lapatinib	Cheng et al.,2017
	TPC-1	Human	CCDC6-RET (RET/PTC1) gene fusion, Homozygous for CDKN2A p.Ala68fs, Heterozygous for STAG2 p.Gln1089Ter, Heterozygous for TERT c.228C >T	Bcl2 family inhibitor/MAPK inhibitors	ABT-737 + PLX4720 + PD32590	Gunda et al., 2017
				Raf inhibitor/immunotherapy	PLX4720 + anti PDL-1	Brauner et al., 2015
				MEK inhibitor/dietary supplement	U0126 + sodium selenite	Kim et al., 2020
				BRAF inhibitor + EZH2 inhibitor	Selumetinib/dabrafenib + tazemetostat	Fu et al., 2020
	TBP-3868	Murine	BrafV600E/WT; p53−/−	SRC inhibitor/Raf inhibitor	Dasatinib + PLX4720	Vanden Borre et al., 2014
Follicular form	FTC-133	Human	Homozygous for FLCN p.His429fs (c.1285delC), Homozygous for MSH6 p.Lys1045fs (c.3135delG),Homozygous for NF1 p.Cys167Ter (c.501T >A), Homozygous for PTEN p.Arg130Ter (c.388C >T), Homozygous for TERT c.228C >T (-124C >T), Homozygous for TP53 p.Arg273His (c.818G >A)	MEK1/2 inhibitor/mTOR inhibitor	RDEA119 + temsirolimus	Liu et al., 2010
				Raf inhibitor/PI3K inhibitor	RAF265 + Dactolisib (BEZ-235)	Jin et al., 2011
				PI3K inhibitor/p53 reactivator	NVP-BKM120 + PRIMA-1Met	Li et al., 2018
				VEGF inhibitor/PI3K inhibitors	Sorafenib+ Dactolisib (BEZ235)/small interfering RNA (siRNA) directed against AKT	Yi H et al.,2017
				VEGF inhibitor/spice	Sorafenib + curcumin	Zhang et al., 2015
				VEGF inhibitor/antimalaric drug	Sorafenib + chloroquine	Yi H et al.,2018
				anti parasitic drug/chemotherapy	Atovaquone + Doxorubicin	Zhuo Lv et al., 2018
	WRO-82-1	Human	BRAF p.Val600Glu (c.1799T >A), TP53 p.Pro223Leu (c.668C >T)	MEK1/2 inhibitor/mTOR inhibitor	RDEA119 + temsirolimus	Liu et al., 2010
				VEGF inhibitor/chemotherapy	Pazopanib + Paclitaxel	Isham et al., 2013
Medullary form	TT	Human	Heterozygous for RET p.Cys634Trp (c.1902C >G), Heterozygous for TBX3 p.Trp197Ter (c.591G >A)	PI3K inhibitor/Raf inhibitor	ZSTK474+ RAF265	Bertazza et al., 2015
				VEGF inhibitor/MEK inhibitor- mTOR inhibitor/MEK inhibitor	Sorafenib + AZD6244/Everolimus + AZD6244	Koh et al., 2012
	MZ-CRC-1	Human	Heterozygous for HIST3H3 p.Arg3Ter (c.7C >T),Homozygous for MAX c.295+1G >A, Homozygous for PBRM1 p.Arg534Ter (c.1600C >T),Heterozygous for RET p.Met918Thr (c.2753T >C)	VEGF inhibitor/MEK inhibitor- mTOR inhibitor/MEK inhibitor	Sorafenib + AZD6244/Everolimus + AZD6244	Koh et al., 2012
				VEGF inhibitor/chemotherapy	Sunitinib + cisplatin	Lopergolo et al., 2014

**Table 2 cells-09-00830-t002:** Details about clinical trials enrolled at clinicaltrials.com, which include combinatorial therapies in thyroid cancer.

N° NCT	Study	Phase	Status	Drugs	Chemotherapies	MAPK Inhibitors	mTOR Inhibitors	VEGF Inhibitors	EGFR Inhibitors	PDL1 Inhibitors	Others
NCT00923247	A Targeted Phase I/II Trial of ZD6474 (Vandetanib; ZACTIMA) Plus the Proteasome Inhibitor, Bortezomib (Velcade), in Adults with Solid Tumors With a Focus on Hereditary or Sporadic, Locally Advanced or Metastatic Medullary Thyroid Cancer (MTC)	I-II	Terminated (Terminated due to slow accrual, primary endpoint reached and investigator left NIH.)	Bortezomib							X
				Vandetanib				X			
NCT01270321	Pasireotide and Everolimus in Adult Patients with Radioiodine-Refractory Differentiated and Medullary Thyroid Cancer	II	Active, not recruiting	Everolimus			X				
				Pasireotide							X
NCT01141309	Evaluating the Combination of Everolimus and Sorafenib in the Treatment of Thyroid Cancer	II	Active, not recruiting	Sorafenib				X			
				Everolimus			X				
NCT02472080	Gemcitabine-Oxaliplatin for Advanced Refractory Thyroid Cancer Patients: A Phase II Study	II	Recruiting	Gemcitabine	X						
				Oxaliplatin	X						
NCT03300765	Trail Evaluating Apatinib With IMRT for Inoperable or Iodine Refractory Thyroid Cancer	II	Recruiting	Apatinib				X			
				Radiation: Intensity modulated radiation therapy							X
NCT01947023	Dabrafenib and Lapatinib Ditosylate in Treating Patients with Refractory Thyroid Cancer That Cannot Be Removed by Surgery	I	Active, not recruiting	Dabrafenib		X					
				Lapatinib Ditosylate					X		
NCT01723202	Dabrafenib With or Without Trametinib in Treating Patients with Recurrent Thyroid Cancer	II	Active, not recruiting	Dabrafenib		X					
				Trametinib		X					
NCT02152995	Trametinib in Increasing Tumoral Iodine Incorporation in Patients with Recurrent or Metastatic Thyroid Cancer	II	Recruiting	Trametinib		X					
				Radiation: Iodine I 124							X
				Radiation: Iodine I-131							X
NCT03065387	Study of the Pan-ERBB Inhibitor Neratinib Given in Combination with Everolimus, Palbociclib or Trametinib in Advanced Cancer Subjects With EGFR Mutation/Amplification, HER2 Mutation/Amplification, HER3/4 Mutation or KRAS Mutation	I	Recruiting	Neratinib					X		
				Everolimus			X				
				Palbociclib							X
				Trametinib		X					
NCT01552434	Bevacizumab and Temsirolimus Alone or in Combination with Valproic Acid or Cetuximab in Treating Patients With Advanced or Metastatic Malignancy or Other Benign Disease	I	Recruiting	Bevacizumab				X			
				Cetuximab					X		
				Temsirolimus			X				
				Valproic Acid							X
NCT03170960	Study of Cabozantinib in Combination with Atezolizumab to Subjects with Locally Advanced or Metastatic Solid Tumors	I-II	Recruiting	Cabozantinib				X			
				Atezolizumab						X	
NCT03085056	Trametinib in Combination with Paclitaxel in the Treatment of Anaplastic Thyroid Cancer	Early I	Recruiting	Trametinib		X					
				Paclitaxel	X						
NCT02152137	Inolitazone Dihydrochloride and Paclitaxel in Treating Patients with Advanced Anaplastic Thyroid Cancer	II	Active, not recruiting	Efatutazone							X
				Paclitaxel	X						
NCT03430882	TAK228 With Carbo and Taxol in Advanced Malignancies	I	Recruiting	Sapanisertib (TAK-228)			X				
				Paclitaxel	X						
				Carboplatin	X						
NCT00077103	Induction Chemotherapy Using Doxorubicin and Cisplatin Followed by Combretastatin A4 Phosphate and Radiation Therapy in Treating Patients with Newly Diagnosed Regionally Advanced Anaplastic Thyroid Cancer	I–II	Terminated (slow accrual)	Filgrastim							X
				Cisplatin	X						
				Doxorubicin hydrochloride	X						
				Fosbretabulin disodium							X
				Radiation							X
NCT00603941	A Phase 1/2 Dose Finding Study of an Experimental New Drug CS7017, an Oral PPARγ Agonist Taken by Mouth Twice Daily in Combination with Paclitaxel Chemotherapy (anaplastic thyroid cancer)	I–II		CS7017							X
				Paclitaxel	X						
NCT03387943	PLD Combined with Cisplatin in the Treatment of Advanced Poorly Differentiated Thyroid Carcinoma	II	Recruiting	Pegylated liposomal doxorubicin hydrochloride (PLD)	X						
				Cisplatin	X						
NCT03181100	Atezolizumab Combinations with Chemotherapy for Anaplastic and Poorly Differentiated Thyroid Carcinomas	II	Recruiting	Nab-paclitaxel	X						
				Paclitaxel	X						
				Vemurafenib		X					
				Cobimetinib		X					
				Atezolizumab						X	
				Bevacizumab				X			
NCT02936102	A Study of FAZ053 Single Agent and in Combination with PDR001 in Patients with Advanced Malignancies. (anaplastic thyroid cancer)	I	Recruiting	FAZ053							X
				PDR001						X	
NCT03122496	Immunotherapy and Stereotactic Body Radiotherapy (SBRT) for Metastatic Anaplastic Thyroid Cancer	I	Recruiting	Durvalumab						X	
				Tremelimumab							X
				Radiation: Stereotactic Body Radiotherapy (SBRT)							X
NCT03211117	Pembrolizumab, Chemotherapy, and Radiation Therapy with or Without Surgery in Treating Patients with Anaplastic Thyroid Cancer	II	Active, not recruiting	Docetaxel	X						
				Doxorubicin Hydrochloride	X						
				Radiation: Intensity-Modulated Radiation Therapy							X
				Pembrolizumab						X	
NCT03360890	Pembrolizumab With Chemotherapy for Poorly Chemoresponsive Thyroid and Salivary Gland Tumors	II	Recruiting	Pembrolizumab						X	
				Docetaxel	X						
NCT03217747	Study to Evaluate the Safety and Tolerability of Avelumab in Combination with Other Anticancer Therapies in Patients with Advanced Malignancies	I-II	Recruiting	Avelumab						X	
				Utomilumab							X
				PF-04518600							X
				Cisplatin	X						
				Radiation							X
NCT03246958	Nivolumab Plus Ipilimumab in Thyroid Cancer	II	Recruiting	Nivolumab						X	
				Ipilimumab							X
NCT03753919	Durvalumab Plus Tremelimumab for the Treatment of Patients with Progressive, Refractory Advanced Thyroid Carcinoma—The DUTHY Trial	II	Not yet recruiting	Durvalumab						X	
				Tremelimumab							X
NCT00354523	Imatinib in Combination with Dacarbazine and Capecitabine in Medullary Thyroid Carcinoma	I	Terminated (Study closed following Phase I portion, insufficient activity to continue to Phase II.)	Capecitabine (Xeloda)	X						
				DTIC-Dome (Dacarbazine)							
				Gleevec (Imatinib Mesylate)							X
NCT03215095	RAI Plus Immunotherapy for Recurrent/Metastatic Thyroid Cancers	Early I	Recruiting	Durvalumab (Medi4736)						X	
				Radiation							X
NCT03732495	Study of the Efficacy of Lenvatinib Combined with Denosumab in the Treatment of Patients with Predominant Bone Metastatic Radioiodine Refractory Differentiated Thyroid Carcinomas	II	Not yet recruiting	Lenvatinib				X			
				Denosumab							X
NCT02973997	Lenvatinib and Pembrolizumab in DTC	II	Recruiting	Lenvatinib				X			
NCT03506048	Lenvatinib and Iodine Therapy in Treating Patients with Radioactive Iodine-Sensitive Differentiated Thyroid Cancer	II	Not yet recruiting	Lenvatinib				X			
				Radiation: iodine I 131							
NCT02432274	Study of Lenvatinib in Children and Adolescents with Refractory or Relapsed Solid Malignancies and Young Adults with Osteosarcoma	I-II	Recruiting	Lenvatinib				X			
				Ifosfamide	X						
				Etoposide	X						
NCT02393690	Iodine I-131 With or Without Selumetinib in Treating Patients with Recurrent or Metastatic Thyroid Cancer	II	Recruiting	Selumetinib		X					
				Radiation: Iodine I-131							X
NCT03647657	177Lu-PP-F11N in Combination with Sacubitril for Receptor Targeted Therapy and Imaging of Metastatic Thyroid Cancer	Early I	Not yet recruiting	177Lu-PP-F11N							X
				Sacuitril							X

**Table 3 cells-09-00830-t003:** Details about closed clinical trials enrolled at clinicaltrials.com, which include combinatorial therapies in thyroid cancer.

Study	Phase	Molecules and Dosage	Patients Treated by Combination	Adverse Effects	Median Progression Free Survival
The antiproliferative effect of pasireotide LAR alone and in combination with everolimus in patients with medullary thyroid cancer: a single-center, open-label, phase II, proof-of-concepts study Faggiano et al., 2018	II	Pasireotide (SOM230): 60 mg IM every 28 ± 2 days Everolimus (RAD001): 10 mg per os/day	7	Hyperglycemia Fatigue Dyspnoea	9.0 months
Phase I/II trial of Vandetanib and Bortezomib in Adults with Locally Advanced or Metastatic Medullary Thyroid cancer De Rivero et al., 2018	I (II)	Bortezomib: 1.3 mg/m^2^ IV on days 1, 4, 8, and 11 Vandetanib: 300 mg orally/day	19	Hypertension Fatigue Thrombocytopenia Diarrhea Arthralgia	X
Valproic acid, a Histone Deacetylase Inhibitor, in Combination with Paclitaxel for Anaplastic Thyroid cancer: Results of a Multicenter Randomized Controlled Phase II/III trial Catalano et al., 2016	II/III	Valproic acid: 1000 mg/day Paclitaxel: IV 80 mg/m^2^/week	11	Hematologic toxicity Gastrointestinal toxicity Neurotoxicity Cardiac toxicity Muscle and skeletal toxicity	4 months
Efatutazone, an oral PPAR-γ Agonist, in Combination with Paclitaxel in Anaplastic Thyroid Cancer: Results of a multicenter Phase 1 Trial Smallridge et al., 2013	I	Efatutazone: 0.15 mg/0.3 mg/0.5 mg twice/day Paclitaxel: every 3 weeks	15	Anemia Oedema	3 months for 0.15 mg 4.5 months for 0.3 mg
Combination of Temsirolimus and Sorafenib in the Treatment of Radioactive Iodine Refractory Thyroid Cancer Sherman et al., 2017	II	Temsirolimus: 25 mg IV weekly Sorafenib: 200 mg orally twice/day 8 weeks of treatment	37	Anemia Nausea Vomiting Hypertension	Partial Response 8 (26.7%) Stable Disease 21 (70.0%) Disease Progression 1 (3.3%)
Dabrafenib and trametinib treatment in patients with locally advanced or metastatic BRAF V600E Mutant Anaplastic Thyroid cancer Subbiah et al., 2017	II	Dabrafenib: 150 mg twice/day Trametinib: 2 mg once/day	16	Fatigue Pyrexia Nausea	X
Randomized Safety and Efficacy Study of Fosbretabulin (CA4P) with Paclitaxel/Carboplatin Against Anaplastic Thyroid Carcinoma Sosa et al., 2014	II-III	CA4P: 60 mg/m^2^ on days 1, 8, and 15 Paclitaxel: 200 mg/m^2^ on Day 2 Carboplatin: AUC 6	55	Bronchitis Anemia Leukopenia Diarrhea Dysphagia Vomiting Fatigue Dyspnea Alopecia Hypertension	5.2 months
A Phase I/II Trial of Crolibulin (EPC2407) Plus Cisplatin in Adults with Solid Tumors With a Focus on Anaplastic Thyroid Cancer (ATC) Gramza et al., National Cancer Institute, National Institutes of Health, Bethesda, MD	I-II The phase II portion was not completed because it was impossible to recruit.	Crolibulin: IV 8–20 mg/m^2^ Cisplatin: IV 75–100 mg/m^2^	26	Nausea Pancreatitis Fatigue Hypoalbuminemia Hypomagnesemia Hyponatremia Hypertension Platelet count decreased	X
A phase I study of imatinib, dacarbazine, and capecitabine in advanced endocrine cancers Halperin et al., 2014	I	Capecitabine (Xeloda): 500 mg/m^2^ twice/day on days 1–14 DTIC-Dome (Dacarbazine): 251 mg/m^2^/day on days 1–3 Gleevec (Imatinib Mesylate): 300 mg/day on days 1–21	8	Constipation Diarrhea Dyspnea Oedema Fatigue Insomnia Nausea Pain	X
A phase I and pharmacokinetic study of irofulven and capecitabine administered every 2 weeks in patients with advanced solid tumors Alexandre et al., 2007	I	Irofulven: IV 0.4 mg over 30 min on days 1 and 15 every 4 weeks Capecitabine: 2000 mg/m^2^/day on days 1–15	4	Leukopenia Neutropenia Anemia Thrombocytopenia	X
A phase I study of pazopanib in combination with escalating doses of 131 I in patients with well-differentiated thyroid carcinoma borderline refractory to radioiodine Chow et al., 2017	I	Pazopanib hydrochloride: 800 mg/day ≥600 mg/day Radiation: iodine I 131	6	Fatigue Anorexia Diarrhea Dysgeusia Nausea Transaminemia Hypertension Thrombocytopenia	6.7 months

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
