# Peer review of "Combinatorial Therapies in Thyroid Cancer: An Overview of Preclinical and Clinical Progresses"

_cells, 2020, doi:10.3390/cells9040830_

Round 1

Reviewer 1 Report

The review by Gheysen et al. covers an enormous amount of data regarding advanced thyroid cancer either preclinical or clinical. The risk of such reviews is also that much is missed.

In the introduction a general overview of (recurrent) thyroid cancer is given. Also categories (misspelled on line 66) of patients are described. As a more general knowledge the authors do not address the fact that for differentiated non-medullary TC (DTC, NMTC) the recurrent cases are either PTC with BRAFV600E DNA variations or Hurthlecel carcinomas. In fact Hurthle cell carcinomas (HCC) are not mentioned at all. HCC is nowadays been considered as an entity on its own and does not form a subgroup of follicular thyroid neoplasm. ATC can have different DTC precursors among which BRAF positive PTC and ATC. Please see current literature.

As far as the following chapters 2 and 3 it is difficult to digest all the data and these data are in fact not complete. Why the authors only focus on combination therapies is in fact unclear to me. For instance the recent development of targeting gene fusions in a tumor agnostic manor gives very promising results also in thyroid cancer (e.g. targeting NTRK1, 2 or 3).        

Author Response

We thank this reviewer whose comments help us to improve our manuscript.

Our work indeed covers the most significant preclinical and clinical data and presents 3 large Tables to summarize them giving a complete overview of the literature. The reported data are complete based on the most relevant ones.

Also, we focus on combination therapies to limit the field of the review on this topic which is very quickly evolving. Indeed, the treatment of advanced thyroid cancer has undergone rapid evolution in the last decade, with multiple kinase inhibitor drug approvals for each subtype of thyroid cancer.

As addressed by the Reviewer, HCC is now described in the introduction as an additional form of well-differentiated thyroid cancer. We report that HCC is only detected in about 3% of patients but it is not considered a subtype of FTC in the 2017 WHO Classification of Tumors of Endocrine Organs (Janovitz, Endocr Pathol 29, 2018). Also, as PTC and FTC, HCC has a good prognosis (Schatz-Siemers, Diagn Cytopathol 47, 2019). The tumorigenesis process of the anaplastic form has been addressed in the paragraph devoted to the anaplastic form.

The Table 3 summarizes the most pertinent data describing the study phase, the molecules and doses, the patient numbers, the side effects and the median PFS. Moreover, the median PFS is now presented in months in the table.

Finally, gene fusion alterations are now described in a new section entitled “targeting gene rearrangements in thyroid cancer”, reporting recent data on this additional and promising therapeutic option inhibiting NTRKs.

We hope that modifications integrated in the document will complete the deficiencies noted by this reviewer.

Reviewer 2 Report

I applaud the authors for their comprehensive review of such a broad topic. I have a few, minor comments:

(1) Some long, narrative paragraphs are difficult to read. The authors may revise the structure of the text to make it more readable.

(2) A few references are not in the standard format, e.g. #7, #14, and #15. Need thorough editing.

(3) The discussion on lenvatinib is relatively omitted. It should be also listed in "2.2.1. VEGF inhibitor used as single agent". Some related works are not included, such as

lenvatinib-golvatinib combination published in Cancer Science (2015) DOI:10.1111/cas.12581

lenvatinib-HNHA combination published in Neoplasia (2018) DOI:10.1016/j.neo.2017.12.003

(4) m2 as a unit of the body surface area should be superscript (line 664 and many others).

(5) Some references seem misplaced and irrelevant to the text, e.g.
29 (line 97), 38 (line 138), 39 (line 140)

Author Response

Firstly, we would like to thank this reviewer for his kind look and we tried to do our best to improve the manuscript regardless of the comments.

Some paragraphs in the part Combination to MAPK/MEK inhibitors have been subdivided for a better fluidity during the reading.

Some references were indeed not correctly insert in the standard format but are now corrected.

A discussion about the molecule lenvatinib has been addressed and the publications mentioned by the reviewer added in the text (561-580).

The unit of body surface has been corrected with a superscript.

Finally, references in the text have been checked.

We hope that modifications applied to the manuscript will satisfy the comments of this reviewer.

Round 2

Reviewer 1 Report

The authors adapted the manuscript as requested. The message that Hurthle cell carcinoma (HCC) is infrequent is correct in consecutive series. However in recurrent cases a large proportion consists of HCC due to the loss of NaI transport. 

See van der Tuin et al. Eur J Endocrinol 2019 PubMed PMID 30668525

Furthermore it is not only NTRK1,2,3 but also ALK gene fusions that are targetable in this context. 

Author Response

Firstly, we would like to thank the reviewer for his expert view especially on the part speaking of gene fusions. We have expanded this part as asked by the reviewer.

We hope that these modifications will satisfy the reviewer.